  

# Metabolic switch and epithelial–mesenchymal transition cooperate to regulate pluripotency

Hao Sun[1,2,3,4,†] , Xiao Yang[1,2,3,4,†], Lining Liang[1,2,3,4,†], Mengdan Zhang[1,2,3,4,†], Yuan Li[1,2,3],
Jinlong Chen[1,2,3,4] , Fuhui Wang[1,2,3,4], Tingting Yang[1,2,3,4], Fei Meng[1,2,3,4], Xiaowei Lai[1,2,3,4],
Changpeng Li[1,2,3], Jingcai He[1,2,3], Meiai He[1,2,3,4], Qiaoran Xu[1,2,3,4], Qian Li[1], Lilong Lin[1,2,3,4],
Duanqing Pei[1,2,3,4,5,*] & Hui Zheng[1,2,3,4,5,**]

## Abstract

Both metabolic switch from oxidative phosphorylation to glycolysis (OGS) and epithelial–mesenchymal transition (EMT) promote cellular reprogramming at early stages. However, their connections have not been elucidated. Here, when a chemically defined medium was used to induce early EMT during mouse reprogramming, a facilitated OGS was also observed at the same time. Additional investigations suggested that the two events formed a positive feedback loop via transcriptional activation, cooperated to upregulate epigenetic factors such as Bmi1, Ctcf, Ezh2, Kdm2b, and Wdr5, and accelerated pluripotency induction at the early stage. However, at late stages, by over-inducing glycolysis and preventing the necessary mesenchymal–epithelial transition, the two events trapped the cells at a new pluripotency state between naïve and primed states and inhibited further reprogramming toward the naïve state. In addition, the pluripotent stem cells at the new state have high similarity to epiblasts from E4.5 and E5.5 embryos, and have distinct characteristics from the previously reported epiblast-like or formative states. Therefore, the time-dependent cooperation between OGS and EMT in regulating pluripotency should extend our understanding of related fields.

**Keywords** mesenchymal–epithelial transition; metabolic switch; pluripotent state; reprogramming
**Subject Categories** Chromatin, Transcription & Genomics; Metabolism; Stem Cells & Regenerative Medicine
**The EMBO Journal (2020) 39: e102961**

## Introduction

The generation of induced pluripotent stem cells (iPSCs) with *Oct4*, *Klf4*, *c-Myc,* and *Sox2* (OKMS) is a complex process (Takahashi & Yamanaka, 2006) involving changes in multiple aspects such as epigenetic status (Koche *et al*, 2011) and cellular metabolism (Folmes *et al*, 2011). The metabolic switch from oxidative phosphorylation (OXPHOS) to glycolysis (OGS) is a major change during reprogramming (Folmes *et al*, 2011). By upregulating *Pdk1-3* and *Pkm2*, HIF1α promotes OGS and accelerates reprogramming (Prigione *et al*, 2014). In addition, although HIF1α promotes reprogramming during the whole reprogramming process, stabilization of HIF2α during late stages represses reprogramming (Mathieu *et al*, 2014). Such time-dependent effects of HIF2α are similar to the effects of epithelial–mesenchymal transition (EMT) during somatic cell reprogramming (Liu *et al*, 2013).

Mesenchymal–epithelial transition (MET) has been revealed as an early and necessary event during the reprogramming of mouse embryonic fibroblasts (MEFs) (Li *et al*, 2010; Samavarchi-Tehrani *et al*, 2010). Moreover, a sequential introduction of the four Yamanaka factors (*Oct4* and *Klf4* were delivered first, *c-Myc* next, and *Sox2* last, OK + M + S) results in a temporal EMT before the conventional MET and promotes reprogramming (Liu *et al*, 2013). The beneficial roles of sequential EMT-MET or early EMT have also been confirmed in B-cell reprogramming with CCAAT/enhancer-binding protein-α and the four Yamanaka factors (*Oct4*, *Klf4*, *c-Myc,* and *Sox2* were delivered simultaneously, OKMS) (Di Stefano *et al*, 2014). However, EMT induced at the late stage of reprogramming significantly impaired the induction of pluripotency (Liu *et al*, 2013).

The similarity between the time-dependent effects of EMT and HIF2α suggests a potential cooperation between EMT and OGS. In addition, both sequential EMT-MET and metabolic switch

1   CAS Key Laboratory of Regenerative Biology, Joint School of Life Sciences, Guangzhou Institutes of Biomedicine and Health, Chinese Academy of Sciences, Guangzhou Medical University, Guangzhou, China
2   Guangzhou Regenerative Medicine and Health Guangdong Laboratory, Guangzhou, China
3   Guangdong Provincial Key Laboratory of Stem Cell and Regenerative Medicine, Guangzhou, China
4   University of Chinese Academy of Sciences, Beijing, China
5   Institutes for Stem Cell and Regeneration, Chinese Academy of Sciences, Beijing, China
    *Corresponding author. Tel: +86 20 32015334; Fax: +86 20 32015231; E-mail: pei_duanqing@gibh.ac.cn
    **Corresponding author. Tel: +86 20 32015334; Fax: +86 20 32015231; E-mail: zheng_hui@gibh.ac.cn
    †These authors contributed equally to this work

play essential roles during embryonic development and cancer development (Chaffer *et al*, 2007; Thiery *et al*, 2009; Folmes *et al*, 2012), which further supports the cooperation between EMT and OGS.

Mouse embryonic stem cells (ESCs) are derived from the inner cell mass (ICM) of blastocysts, described as the naïve state, and rely on LIF-STAT3 signaling to maintain pluripotency. Mouse epiblast stem cells (EpiSCs) are isolated from the postimplantation epiblast, referred to as the primed state, depending on bFGF/ activin signaling. These two types of PSCs differ from each other when metabolic characteristics or the expression of epithelial/ mesenchymal markers are compared (Folmes *et al*, 2012; Weinberger *et al*, 2016). Generally, during the transition from naïve to primed PSCs, both EMT and OGS can be observed. Thus, it is reasonable to suggest cooperation between EMT and OGS during the regulation of pluripotency.

Sequential EMT-MET can be induced in MEFs with a new chemically defined medium (5C medium), which includes DMEM/F12 (1:1), N2 supplement (1%), basic fibroblast growth factor (bFGF, 20 ng/ml), β-mercaptoethanol (55 μM), vitamin C (Vc, 55 μg/ml), and leukemia inhibitory factor (LIF, 1,000 unit/ml) (He *et al*, 2015, 2017). Therefore, in the current study, we first sought to determine whether and how the 5C medium influences the reprogramming of MEFs by investigating OGS and EMT. In addition, the generated iPSCs were compared with PSCs in naïve and primed states to explore the underlying mechanisms.

# Results

## 5C medium promotes reprogramming by facilitating both early EMT and OGS

The chemically defined 5C medium (Dataset EV1) was used after simultaneously introducing *Oct4*, *Klf4*, *c-Myc*, and *Sox2* into OG2 transgenic MEFs (*egfp* reporter genes driven by the *Oct4* promoter, GOF18ΔPE) (Szabo *et al*, 2002). 5C medium induced approximately twofold more Oct4GFP$^+$ colonies than conventional mES medium (Fig 1A and B). Consistently, when the expression of key pluripotency markers was monitored and compared during the whole reprogramming process, 5C medium was found to induce quicker and larger upregulation of these markers than mES medium (Fig 1C). Oct4GFP$^+$ cells generated with 5C medium (5C-Oct4GFP$^+$) had typical characteristics of PSCs including silenced expression of exogenous factors, significant NANOG and REX1 protein levels, and demethylation of the *Oct4* and *Nanog* promoters (Appendix Fig S1).

The ability of 5C medium to induce sequential EMT-MET was then determined. The expression of several mesenchymal and epithelial markers was monitored and compared during the whole reprogramming process (Fig EV1A). 5C medium increased the expression of mesenchymal markers but suppressed the expression of epithelial markers on day 3 (Fig EV1A), which suggests that 5C medium does induce EMT at the early stage. In addition, these expression changes were reversed on day 6 and thereafter, which suggested a late MET. On the other hand, mES medium induced MET as early as on day 3 (Fig EV1A). Collective analysis with the expression of mesenchymal and pluripotency markers confirmed

sequential EMT-MET and early pluripotency induction during reprogramming with 5C medium (Fig EV1B).

In addition, since epithelial and mesenchymal cells differ a lot in their migration abilities, cell migration can be used to quantify EMT or MET. Live-cell imaging was used to determine the migration ability of cells and further confirmed sequential EMT-MET during reprogramming with 5C medium (Fig EV1C).

Another question was whether sequential EMT-MET or early EMT contributes to the promoted reprogramming. Cells on day 3 during reprogramming were grouped based on their migration abilities, and their cell fates were traced via a live-cell imaging system. The cells with low or medium migration abilities generated more Oct4GFP$^+$ colonies with mES medium (Fig EV1D). However, the cells with medium or high migration abilities generated more Oct4GFP$^+$ colonies with 5C medium (Fig EV1D).

TGFβ (TGFβ1/2/3 1 ng/ml each) and RepSox (1 μM), a TGF pathway inhibitor (Ichida *et al*, 2009), were then used to modulate early EMT as reported previously (Liu *et al*, 2013) (Fig EV1E). When RepSox was used on days 2–7 during reprogramming with 5C medium to block early EMT (Fig EV1F and G), the generation of iPSCs was inhibited (Fig EV1H). When TGFβ was used on days 2–7 during reprogramming with mES medium to induce early EMT (Fig EV1F and G), significantly more Oct4GFP$^+$ colonies were generated (Fig EV1H). Therefore, 5C medium promotes reprogramming by inducing early EMT at least partially.

Then, the question arises as to whether OGS was also influenced during the reprogramming with 5C medium. The metabolic characteristics of the cells were determined on day 6 during reprogramming with the Seahorse instrument (Fig 1D and E). Increase in extracellular acidification rate (ECAR) after adding glucose indicated the level of glycolysis of the cells, while a decrease in oxygen consumption rate (OCR) after adding oligomycin indicated the level of ATP production or OXPHOS of the cells. 5C medium induced glycolysis to a higher level than mES medium and inhibited OXPHOS to a lower level than mES medium (Fig 1D and E). Thus, 5C medium induced a larger metabolic switch from OXPHOS to glycolysis than mES medium. The larger OGS caused by 5C medium was further confirmed by the expression changes of glycolysis markers and the activity of HIF1α during reprogramming (Appendix Fig S2A and B).

By modulating the expression of *Hif1α* with a retrovirus system (Appendix Fig S2C), we were able to control the activity of HIF1α and subsequently regulate energy metabolism (Fig 1F). Exogenous expression of *Hif1α* promoted glycolysis and impaired OXPHOS, while suppressing *Hif1α* expression with sh-RNA led to the opposite results (Appendix Fig S2D and E). Consistent with previous reports (Mathieu *et al*, 2014; Prigione *et al*, 2014), increased HIF1α activity and facilitated OGS played beneficial roles during reprogramming (Fig 1G).

Oxidative phosphorylation to glycolysis during reprogramming was further modulated by controlling the expression of *Pdk1/2* or by using small-molecule compounds, such as oligomycin (to impair oxidative phosphorylation) or 2-deoxy-D-glucose (2-DG, to inhibit glycolysis; Fig 1H and Appendix Fig S2C). These methods successfully modulated energy metabolism during reprogramming (Appendix Fig S2F and G). Consistent effects were observed during reprogramming with mES and 5C medium. When OGS was further facilitated by overexpressing *Pdk1/2* or using oligomycin during

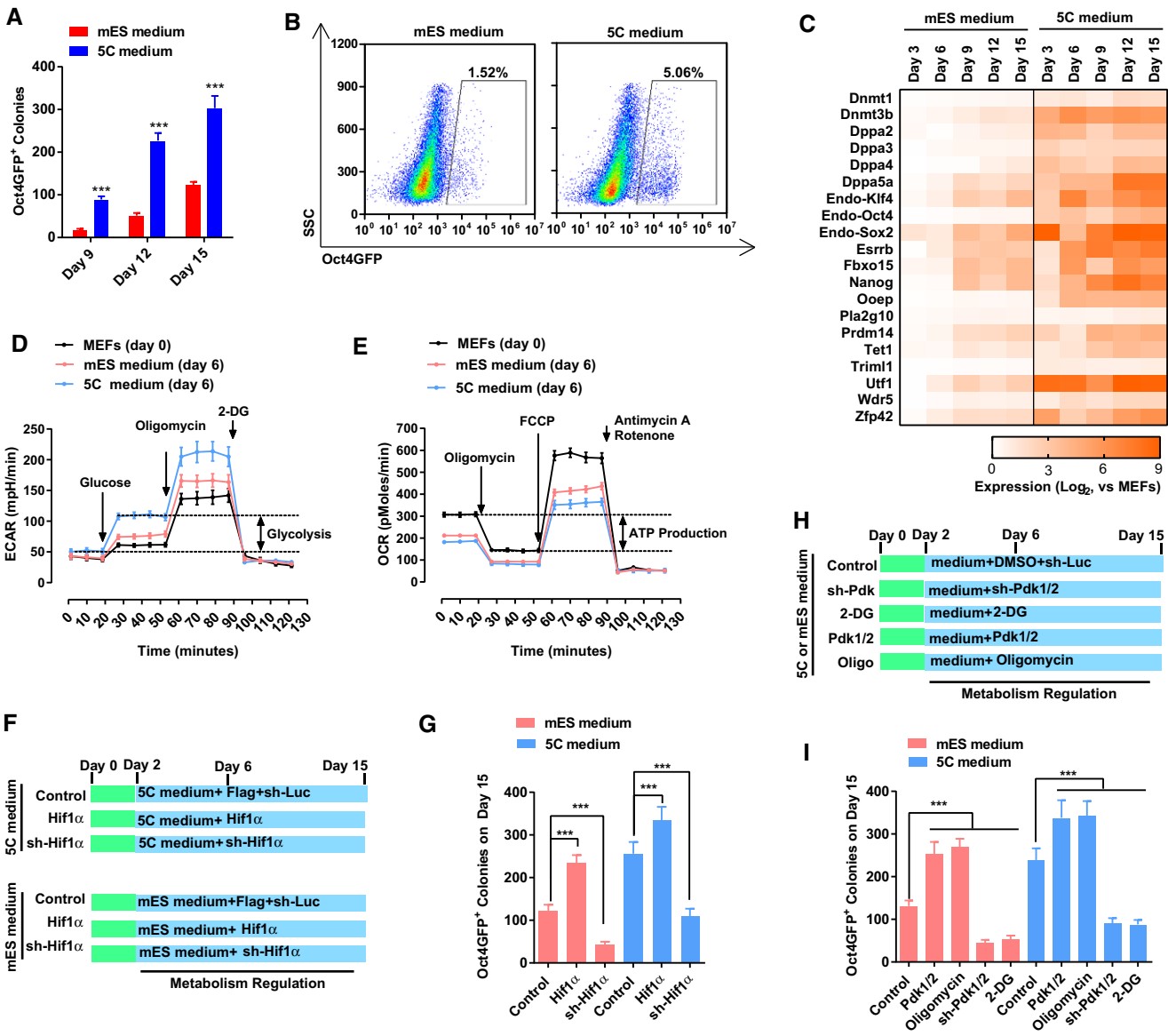

**Figure 1. Facilitated OGS promotes reprogramming.**

A, B 5C or mES medium was used during reprogramming. On day 15, Oct4GFP+ colonies were counted (A). The percentages of Oct4GFP+ cells were also determined with FACS (B).

C The expression of several core pluripotency markers was determined with qPCR at different time points during reprogramming and normalized against those in MEFs.

D, E Energy metabolism was analyzed on day 6 during reprogramming with the Seahorse instrument. Increase in extracellular acidification rate (ECAR) after adding glucose was considered as glycolysis ability of the cells (D), while the decrease in oxygen consumption rate (OCR) was considered as the ATP production ability of the cells (E).

F, G Expression of Hif1α was modulated with overexpression or sh-RNA-mediated knockdown via a retrovirus system during reprogramming (F). The numbers of Oct4GFP+ colonies were determined on day 15 (G).

H, I Expression of Pdk1/2 was modulated with a retrovirus system, and oligomycin (1 μM) and 2-DG (5 mM) were used during reprogramming (H). The numbers of Oct4GFP+ colonies were determined on day 15 (I).

Data information: Experiments were independently repeated at least five times ($n \geq 5$). Error bars represent standard deviations. ***$P < 0.001$. Additional statistical information is listed in Dataset EV7.

reprogramming, more Oct4GFP+ colonies were generated (Fig 1I). When OGS was partially impaired by using sh-RNAs against *Pdk1/2* or 2-DG during reprogramming, reprogramming was significantly inhibited (Fig 1I). Therefore, 5C medium promotes reprogramming by facilitating OGS at least partially.

**5C medium generates less pre-iPSCs by upregulating five epigenetic factors**

During reprogramming with the two media, the generation of pre-iPSCs and iPSCs colonies was determined by counting the

AP⁺Oct4GFP⁻ and AP⁺Oct4GFP⁺ colonies, respectively, on day 15. We found that 5C medium induced many fewer pre-iPSC colonies than mES medium (Fig 2A and B).

These results suggested that 5C medium might promote reprogramming by facilitating the conversation of pre-iPSCs to iPSCs. To test this hypothesis, Oct4GFP⁻ cells generated with mES or 5C medium (mES-Oct4GFP⁻ or 5C-Oct4GFP⁻ cells) were separated with fluorescent-activated cell sorting (FACS) and compared with RNA-Seq (GSE103765 & GSE103791). Genes with differential expression in pre-iPSCs and iPSCs/ESCs were selected (Data ref: Mikkelsen *et al*, 2008; Data ref: Sridharan *et al*, 2009). The expression of these genes in mES-Oct4GFP⁻ and 5C-Oct4GFP⁻ cells is listed in Appendix Fig S3A. 5C-Oct4GFP⁻ cells had closer gene expression profiles to iPSCs or ESCs than mES-Oct4GFP⁻ cells. In addition, when we classified the barrier in gene expression from pre-iPSCs to iPSCs into four different types, 5C medium helped Oct4GFP⁻ cells overcome more barriers than mES medium (Appendix Fig S3B). Therefore, the barriers which trapped cells at pre-iPSC stage during reprogramming with mES medium were significantly overcome by 5C medium.

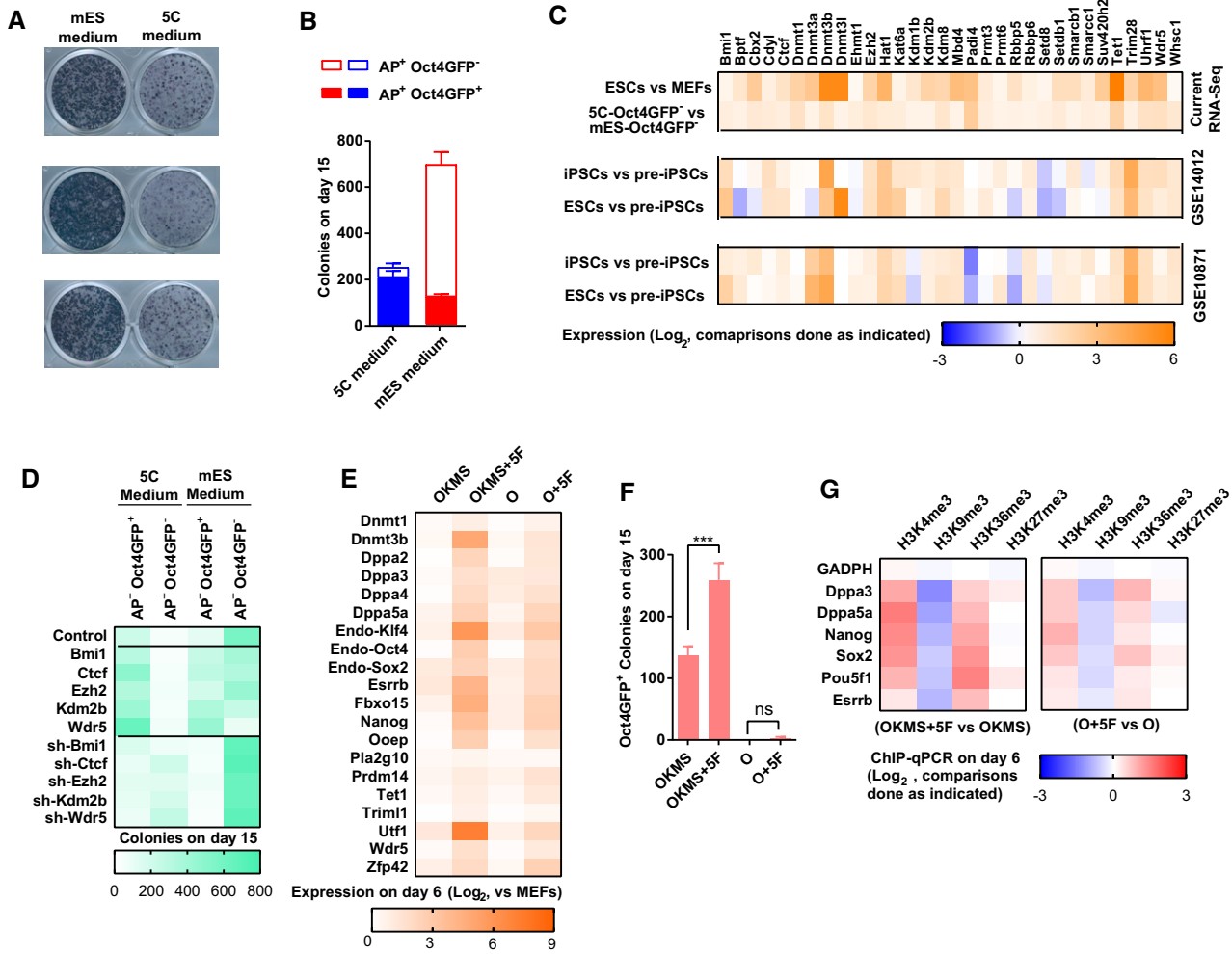

**Figure 2. 5C medium removes epigenetic barriers during reprogramming.**

A, B  5C and mES medium were used during reprogramming. Representative alkaline phosphatase (AP) staining on day 15 was provided in (A). AP⁺Oct4GFP⁻ and AP⁺Oct4GFP⁺ colonies were counted on day 15 (B).

C  Thirty-two epigenetic factors were selected because they had higher expression in 5C-Oct4GFP⁻ than in mES-Oct4GFP⁻. Their expression in ESCs, iPSCs, and pre-iPSCs in two previous reported assays (GSE14012 and GSE10871) was also listed.

D  The expression of *Bmi1*, *Ctcf*, *Ezh2*, *Kdm2b*, and *Wdr5* was modulated with overexpression or sh-RNA-mediated knockdown via a retrovirus system during reprogramming. AP⁺Oct4GFP⁻ and AP⁺Oct4GFP⁺ colonies were counted on day 15.

E–G  During reprogramming with mES medium, *Bmi1*, *Ctcf*, *Ezh2*, *Kdm2b*, and *Wdr5* were overexpressed simultaneously with the four Yamanaka factors (OKMS + 5F) or *Oct4* (O + 5F). All factors were delivered simultaneously via a retrovirus system on days 0 and 1. Reprogramming with only Yamanaka factors (OKMS) or *Oct4* (O) served as control. The expression of pluripotency markers was determined with qPCR on day 6 during reprogramming (E). The numbers of Oct4GFP⁺ colonies were determined on day 15 (F). The histone methylation on core pluripotency loci was determined on day 6 with ChIP-qPCR (G).

Data information: Experiments were independently repeated at least five times (*n* ≥ 5) except for dataset analysis. Error bars represent standard deviations. ***P < 0.001. Additional statistical information is listed in Dataset EV7.

The low expression and high H3K9 methylation of core pluripotency loci in pre-iPSCs are critical barriers for further conversion to iPSCs, which can be overcome by Vc treatment (Chen *et al*, 2012). Because Vc is included in both 5C and mES medium (Dataset EV1), the difference in H3K9 methylation might not explain the current difference of the two media in inducing pre-iPSCs. In addition, the expression of methyltransferases and demethylases targeting H3K9 was similar in 5C- and mES-Oct4GFP⁻ cells (Appendix Fig S3C).

We then hypothesized that 5C medium might help overcome other epigenetic barriers. Thirty-two epigenetic factors with significantly higher expression in 5C-Oct4GFP⁻ cells than in mES-Oct4GFP⁻ cells were identified (Fig 2C). Approximately 60% of these genes had higher expression in ESCs than in MEFs and had higher expression in iPSCs than in pre-iPSCs. Within these genes, *Bmi1*, *Ctcf*, *Ezh2*, *Kdm2b*, and *Wdr5* have been reported to promote reprogramming (Ang *et al*, 2011; Moon *et al*, 2011; Wang *et al*, 2011, 2017; Onder *et al*, 2012). When these five factors were individually tested, overexpression of these genes facilitated reprogramming, while suppression of these genes with sh-RNAs decreased the number of Oct4GFP⁺ colonies on day 15 (Fig 2D and Appendix Fig S3D).

When these five factors were overexpressed simultaneously during reprogramming with mES medium, increase of Oct4GFP⁺ colonies on day 15 and upregulation of pluripotency markers on day 6 were observed (Fig 2E and F). In addition, consistent epigenetic changes (increase in H3K4me3 and H3K26me3, decrease in H3K9me3 and H3K27me3) on several core pluripotency loci were also observed on day 6 with ChIP-qPCR (Fig 2G). Moreover, *Oct4* is sufficient to help these five factors induce similar epigenetic and expression changes (Fig 2E–G). Therefore, 5C medium promotes reprogramming by upregulating these five epigenetic factors at least partially.

### Early EMT and OGS upregulate the five epigenetic factors

We then sought to determine whether early EMT or facilitated OGS contributes to the upregulation of these epigenetic factors. The promoters of *Bmi1*, *Ctcf*, *Ezh2*, *Kdm2b*, and *Wdr5* were analyzed with Pscan software (Zambelli *et al*, 2009). The binding site of HIF1α was enriched on the promoters of all five epigenetic factors (Fig 3A and Dataset EV2). Overexpression of *Hif1α* in MEFs activated the transcription of *Bmi1*, *Ctcf*, *Ezh2*, *Kdm2b*, and *Wdr5* (Fig 3B). The expression of *Hif1α* was then suppressed with sh-RNA at the early stage of reprogramming with 5C medium. Decreases in the transcription of *Bmi1*, *Ctcf*, *Ezh2*, *Kdm2b*, and *Wdr5* were observed (Fig 3C). Therefore, HIF1α-mediated upregulation of these five epigenetic factors is responsible for the abilities of OGS to promote reprogramming.

The binding sites of SNAI2, TWIST1/2, and ZEB1 were also identified on the promoters of the five epigenetic factors (Fig 3A and Dataset EV2). Although the binding sites of SNAI2, TWIST1/2, and ZEB1 were not enriched on all these promoter regions when considered individually, these five promoter regions were enriched with the binding site of at least one of the four key mesenchymal transcriptional factors (Fig 3A). Thus, key transcriptional factors of EMT are potential upstream regulators of these five epigenetic factors.

Overexpression of *Snai2*, *Twist1/2*, or *Zeb1* upregulates the expression of some of the five epigenetic factors during reprogramming with mES medium (Fig 3B). When the expression of *Snai2*, *Twist1/2*, or *Zeb1* was suppressed with sh-RNA at the early stage of reprogramming with 5C medium (Appendix Fig S3E), decreases in the transcription of *Bmi1*, *Ctcf*, *Ezh2*, *Kdm2b*, and *Wdr5* were observed (Fig 3C).

Key transcriptional factors related to OGS and EMT were upstream regulators of *Bmi1*, *Ctcf*, *Ezh2*, *Kdm2b*, and *Wdr5*. Thus, early MET and facilitated OGS may cooperate to upregulate several key epigenetic factors, remove the epigenetic barrier during pluripotency induction, and finally facilitate reprogramming.

### Early EMT and OGS contribute to the less generation of pre-iPSCs

Based on abovementioned results, we hypothesized that early EMT and OGS cooperated to upregulate *Bmi1*, *Ctcf*, *Ezh2*, *Kdm2b*, and *Wdr5*, which subsequently facilitated the conversion from pre-iPSCs to iPSCs. Such a hypothesis explains why 5C medium generates less Oct4GFP⁻ colonies (pre-iPSCs) and more Oct4GFP⁺ colonies (iPSCs) than mES medium.

To confirm the abovementioned hypothesis, the abilities of EMT, OGS, and the five epigenetic factors to facilitate the conversion from pre-iPSCs to iPSCs should be confirmed directly. Pre-iPSC colonies were separated after reprogramming with mES-Vc (mES medium without Vc), mES, or 5C medium (Fig 3D). These pre-iPSC colonies were further cultured for 7 days with mES medium alone or in the presence of other factors, such as HIF1α, TGFβ, or three of the five epigenetic factors (3F, *Ctcf* + *Kdm2b* + *Wdr5*; Fig 3D). HIF1α and TGFβ were used to induce OGS and EMT, respectively. 3F were selected because of their relatively higher abilities to promote reprogramming (Fig 2D) and used to directly mimic their upregulation during reprogramming with 5C medium. mES medium efficiently converted pre-iPSCs separated from reprogramming with mES-Vc medium to iPSCs, while additional factors (HIF1α, TGFβ, or 3F) were required to convert pre-iPSCs separated from reprogramming with mES medium to iPSCs (Fig 3E). Pre-iPSC colonies separated from reprogramming with 5C medium were rare and seldomly converted to iPSCs under the current paradigm (Fig 3E). Therefore, the downstream effects induced by 5C medium can convert the pre-iPSCs generated with mES medium to iPSCs, which explains the higher ability of 5C medium to promote reprogramming.

Then, the abilities of EMT, OGS, and the five epigenetic factors to facilitate the conversion from pre-iPSCs to iPSCs were also confirmed during reprogramming. Similarly, when OGS was facilitated with *Hif1α*, early EMT was induced by TGFβ, or epigenetic changes were mimicked with 3F during reprogramming with mES medium, the number of pre-iPSC (AP⁺Oct4GFP⁻) colonies decreased, while the number of iPSC (AP⁺Oct4GFP⁺) colonies increased (Fig 3F and G). Energy metabolism analysis and cell migration with live-cell imaging on day 6 confirmed the abilities of *Hif1α* and TGFβ to facilitate OGS and EMT, respectively (Fig 3H–J). In addition, the abilities of *Hif1α* and TGFβ to upregulate *Bmi1*, *Ctcf*, *Ezh2*, *Kdm2b*, and *Wdr5* were confirmed with qPCR (Fig 3K). However, overexpression of *Ctcf*, *Kdm2b*, and *Wdr5* (3F) did not affect energy metabolism and cell migration (Fig 3H–K), which further confirmed that the upregulation of these epigenetic factors was downstream of OGS and EMT.

Therefore, consistent with the previous report (Chen *et al*, 2012), Vc helps the cells overcome the H3K9 methylation barrier and

reduces pre-iPSCs. Early EMT, facilitated OGS, and downstream epigenetic factors help the cells overcome additional barriers and accounted for the higher ability of 5C medium to further convert potential pre-iPSCs to iPSCs than mES medium (Fig 3L).

**Facilitated OGS and early EMT form a positive feedback loop**

Surprisingly, when HIF1α was overexpressed, EMT was observed at the early stage of reprogramming (Fig 3H–J). Similarly, when TGFβ

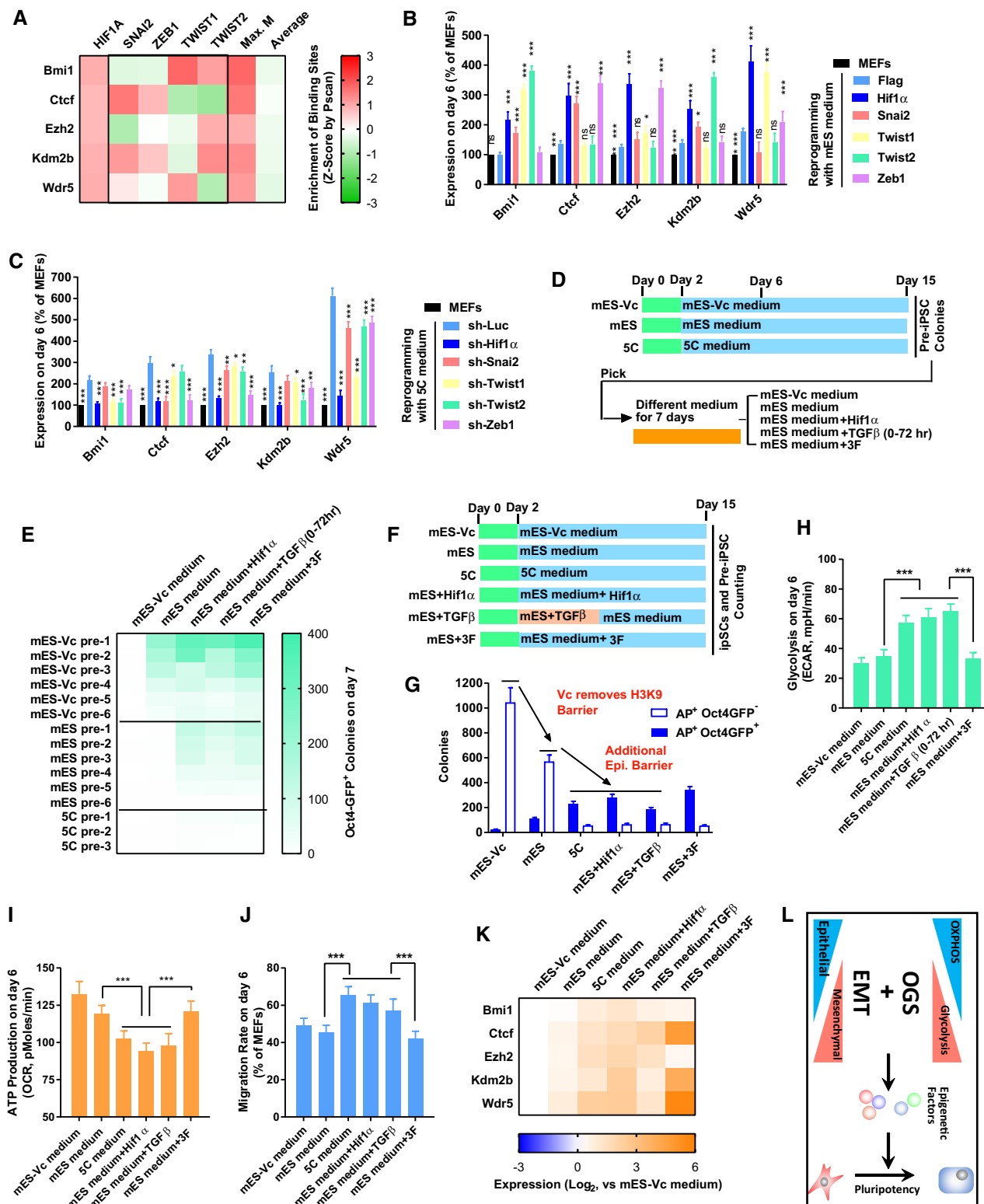

**Figure 3.**

**Figure 3. Early EMT and OGS induce the epigenetic changes.**

A    The enrichment of HIF1α, SNAI2, TWIST1/2, and ZEB1 binding sites on the promoters of *Bmi1*, *Ctcf*, *Ezh2*, *Kdm2b*, and *Wdr5* was determined by Pscan (Zambelli *et al*, 2009). The resulted Z-scores were listed. "Max. M" suggested the maximum Z-score generated with SNAI2, TWIST1/2, and ZEB1 binding sites on one indicated promoter. "Average" suggested the average Z-score of all transcriptional factors tested in the Pscan software on one indicated promoter.

B, C  The expression of *H1f1α*, *Snai2*, *Twist1/2*, and *Zeb1* was modulated with overexpression or sh-RNA-mediated knockdown via a retrovirus system during reprogramming with mES (B) or 5C medium (C). The expression of *Bmi1*, *Ctcf*, *Ezh2*, *Kdm2b*, and *Wdr5* was determined on day 6 with qPCR and normalized against those in MEFs. The comparisons were performed between all groups and corresponding Flag groups.

D, E  pre-iPSCs were isolated during reprogramming with mES-Vc, mES, or 5C medium. These pre-iPSCs were further cultured with a different medium (mES-Vc and mES medium) for 7 days. Additional factors (Hif1α, *Hif1α* overexpression; TGFβ, TGFβ1/2/3 1 ng/ml each; 3F, *Ctcf*, *Kdm2b*, and *Wdr5* overexpression) were used simultaneously with mES medium. The numbers of Oct4GFP⁺ colonies were determined on day 7 (E).

F–K   Reprogramming was also performed with mES-Vc or mES medium. Additional factors (Hif1α, *Hif1α* overexpression; TGFβ, TGFβ1/2/3 1 ng/ml each; 3F, *Ctcf*, *Kdm2b*, and *Wdr5* overexpression) were used simultaneously with mES medium. AP⁺Oct4GFP⁻ and AP⁺Oct4GFP⁺ colonies were counted on day 15 (G). In addition, energy metabolism was determined on day 6 with Seahorse instrument (H and I). Cell migration was determined by measuring the migration rate with live-cell imaging on day 6 (J). The expression of *Bmi1*, *Ctcf*, *Ezh2*, *Kdm2b*, or *Wdr5* was determined on day 6 with qPCR (K).

L    Early EMT and facilitated OGS cooperate to induce pluripotency by removing the additional epigenetic barrier.

Data information: Experiments were independently repeated at least five times ($n \geq 5$) except binding site analysis. Error bars represent standard deviations. *$P < 0.05$, **$P < 0.01$, ***$P < 0.001$. Additional statistical information is listed in Dataset EV7.

was used, OGS was observed (Fig 3H–J). Thus, the connections between OGS and EMT were then investigated.

During reprogramming with mES medium, increasing OGS with *Hif1α* induced early EMT as indicated by changes in the expression of mesenchymal markers and the migration abilities of cells on day 6 (Fig 4A and B). Inhibiting OGS with *sh-Hif1α* impaired early EMT during reprogramming with 5C medium (Fig 4A and B). OGS at the early stage during reprogramming was further modulated by using small-molecule compounds, 2-DG and oligomycin, as in Fig 1H. These methods modulated OGS similarly to *Hif1α* or *sh-Hif1α* but induced EMT to a lesser extent (Fig 4A and B).

To understand why facilitated OGS leads to early EMT, the promoter regions of mesenchymal markers were analyzed. The binding sites of HIF1α had high enrichment on these promoters (Fig EV2A and Dataset EV2). Modulating the expression of *Hif1α* in MEFs resulted in consistent changes of mesenchymal markers (Fig EV2B). Small-molecule compounds, 2-DG and oligomycin, resulted in significant but smaller changes. Therefore, both HIF1α and downstream OGS induce EMT via transcriptional activation of key mesenchymal factors (Fig 4C). In addition, as a transcriptional factor, HIF1α has higher abilities than downstream OGS to induce EMT (Figs 4A–C and EV2A and B).

Next, the influences of early EMT on OGS were studied. *Snai1/2*, *Zeb1/2*, and *Twist1/2* are traditional transcriptional factors that can serve as mesenchymal markers (Thiery, 2002; Thiery *et al*, 2009). The binding sites of at least one of these transcriptional factors are enriched on promoters of glycolysis markers (Fig EV2C and Dataset EV2). In addition, overexpression of these transcriptional factors activated the expression of glycolysis markers in MEFs (Fig EV2D).

Sine EMT at the late stage during reprogramming impaired iPSC generation (Liu *et al*, 2013), overexpression of transcriptional factors was not used to induce EMT during reprogramming. Based on previous research in our laboratory, early EMT was also induced by short-term treatment with TGFβ, inhibitors of lysine-specific histone demethylase 1A (LSD1i, SP2509, 10 nM), and sequential introduction of Yamanaka factors (OK + M + S; Fig 4D) (Liu *et al*, 2013; Sun *et al*, 2016). In the current manuscript, Yamanaka factors were mostly introduced simultaneously via a retrovirus system on day 0 and day 1 during reprogramming, except when "OK + M + S" was used to indicate sequential introduction (Liu *et al*, 2013).

The promoted reprogramming was confirmed by the Oct4GFP⁺ colonies on day 15 and the expression of pluripotency markers on day 6 (Fig EV2E and F). The early EMT induced by these methods was confirmed by cell migration and the expression of mesenchymal markers on day 6 (Fig EV2G and H). Facilitated OGS was observed at the early stage as indicated by energy metabolism studies and the expression of glycolysis markers (Fig 4E–G). Therefore, early EMT can also facilitate OGS via transcriptional activation of key glycolysis factors (Fig 4H). In summary, early EMT and OGS form a positive feedback loop and cooperate to accelerate pluripotency induction (Fig 4I).

Sequential EMT-MET or early EMT is also observed during trans-differentiation from MEFs to functional neurons and differentiation from human ESCs to hepatocytes (He *et al*, 2017; Li *et al*, 2017). Previous RNA-Seq results were analyzed to determine whether similar cooperation existed between early EMT and facilitated OGS in activating the expression of the five epigenetic factors (Data ref: Sun and Zheng, 2016; Data Ref: Hutchins *et al*, 2017). As indicated in Appendix Fig S4A–D, early EMT and facilitated OGS were observed on the first 7 days during a 21-day differentiation from human ESCs to hepatocytes and on the first 5 days during a 16-day transdifferentiation from MEFs to functional neurons. In addition, upregulation of *Bmi1*, *Ctcf*, *Ezh2*, *Kdm2b*, and *Wdr5* was also observed (Appendix Fig S4E and F). Thus, the cooperation between early EMT and OGS is not restricted during reprogramming.

During reprogramming with mES medium, both *Hif1α* and oligomycin induced upregulation of epigenetic factors on day 6 (Appendix Fig S5A). Since oligomycin did not affect the expression of *HIf1α* or *Pdk1/2* (Appendix Fig S5B), and has lower ability than *Hif1α* to induce EMT transcriptional factor (Figs 4A–C and EV2B), the metabolic state contributes to the regulation of epigenetic factors independently of *HIf1α* and EMT transcriptional factors.

In addition, *Hif1α* expression positively correlated with reprogramming efficiency in the presence of 2-DG or oligomycin, OGS or metabolism state also positively correlated with reprogramming efficiency in the context of *Hif1α* overexpression or knockdown (Appendix Fig S5C). Thus, both *Hif1α*-independent OGS and the OGS-unrelated transcriptional activity of HIF1α may contribute to reprogramming. Furthermore, the expression of mesenchymal key transcription was manipulated with TGFβ and RepSox treatment (day 2–7) when *Hif1α* expression and/or OGS were manipulated (Appendix Fig S5D and E). A short treatment of TGFβ consistently

promoted reprogramming. RepSox treatment promoted reprogramming in the presence of *sh-Hif1α* and/or 2-DG, while impaired reprogramming in the presence of *Hif1α* and/or oligomycin, which further explored the connection among *Hif1α* expression, EMT, and OGS. However, because modulating one of the three aspects (*Hif1α* expression, EMT, and OGS) may affect the other two, it is nearly impossible to identify a clear connection between only two of them.

The expression of *Bmi1*, *Ctcf*, *Ezh2*, *Kdm2b*, and *Wdr5* positively correlated with reprogramming efficiency in the presence of 2-DG or

oligomycin (Appendix Fig S5F). Thus, these five factors affect reprogramming efficiency by themselves and also have additive effects.

### A distinct pluripotent state is induced with 5C medium

Because the current MEFs were isolated from transgenic mice that have *egfp* reporter genes driven by the *Oct4* promoter and the distal enhancer (Szabo *et al*, 2002), 5C-Oct4GFP⁺ colonies had distal

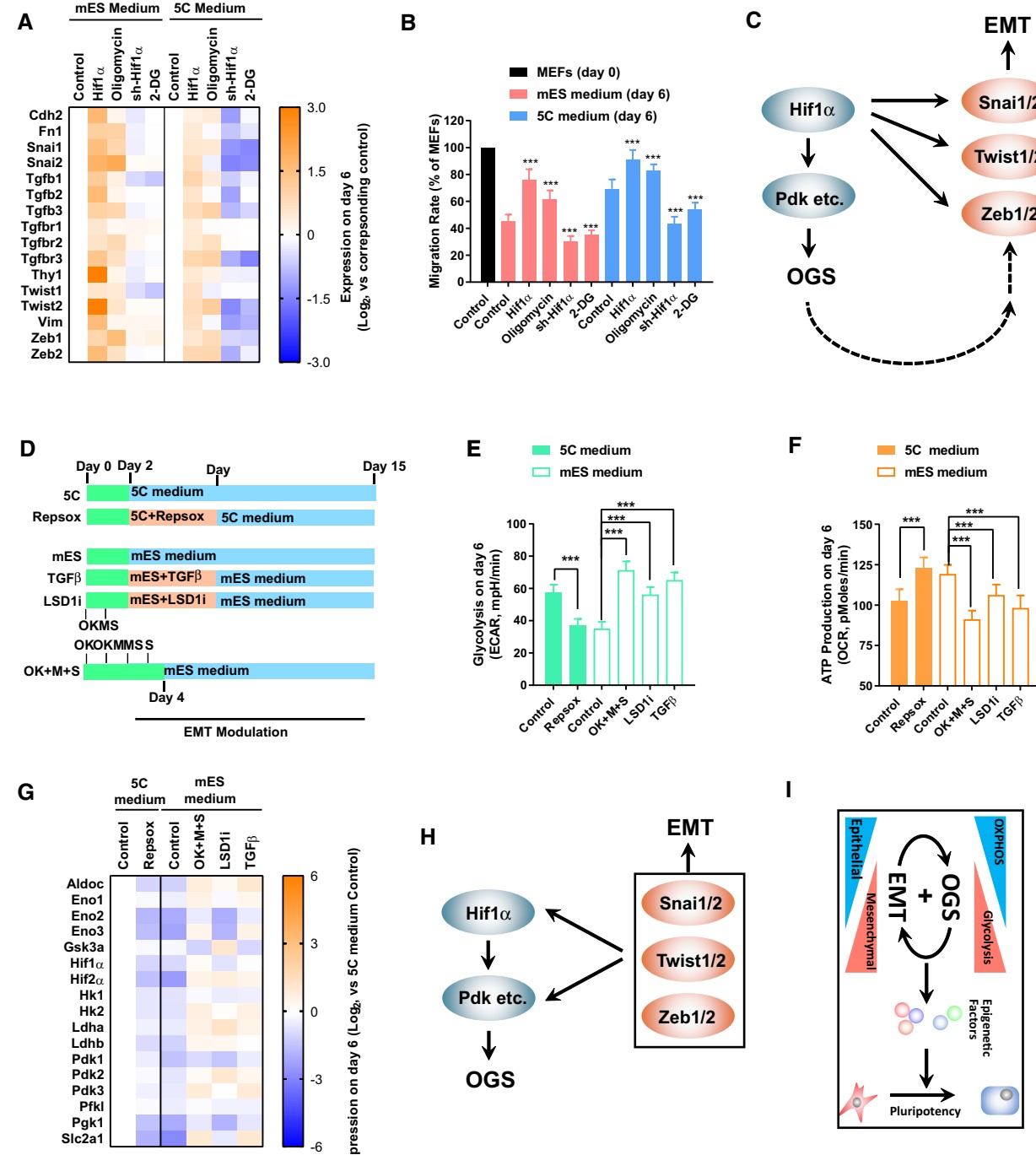

**Figure 4.**

**Figure 4.  Facilitated OGS and early EMT form a positive feedback loop.**

A, B    Expression of *Hif1α* was modulated via a retrovirus system or small-molecule compounds, oligomycin (1 μM) and 2-DG (5 mM), during reprogramming. The expression of mesenchymal markers (A) and cell migration (B) was determined with qPCR and live-cell imaging, respectively, on day 6.

C    HIF1α induces EMT by transcriptionally activating key mesenchymal transcriptional factors, including *Snai1/2*, *Twist1/2*, and *Zeb1/2*. Although to a less extent, inducing OGS directly with *Pdk1/2* or small molecules also activate these factors.

D–G    RepSox (1 μM) was used on days 2–7 to inhibit early EMT during reprogramming with 5C medium. TGFβ (TGFβ1/2/3, 1 ng/ml each) on days 2–7, LSD1i (SP2509, 10 nM), and sequential introduction of Yamanaka factors (OK + M + S) were used to induce early EMT during reprogramming with mES medium (D). Except for OK + M + S group, the four Yamanaka factors were delivered simultaneously. Energy metabolism was determined on day 6 with Seahorse instrument (E and F). The expression of glycolysis markers was determined on day 6 with qPCR (G).

H    Key mesenchymal transcriptional factors, including *Snai1/2*, *Twist1/2*, and *Zeb1/2*, induce OGS by transcriptionally activating *Hif1α* and glycolysis genes.

I    Early EMT and facilitated OGS form a positive feedback loop to induce pluripotency.

Data information: Experiments were independently repeated at least five times ($n \geq 5$). Error bars represent standard deviations. ***$P < 0.001$. Additional statistical information is listed in Dataset EV7.

enhancers of *Oct4* activated and bear naïve characteristics at least partially. However, in contrast to the multilayered morphology of Oct4GFP$^+$ colonies generated with mES medium, 5C-Oct4GFP$^+$ colonies were monolayer (Fig 5A), which suggested the primed characteristic of 5C-Oct4GFP$^+$ colonies (Weinberger *et al*, 2016).

Oct4GFP$^+$ cells generated with the two media, 5C and mES medium, were then sorted out and compared with each other. R1 ESCs cultured in naïve medium served as the control of naïve PSCs (Dataset EV1) (Ying *et al*, 2008). Primed PSCs were generated from R1 ESCs (R1 ESCs-primed) following a previously reported protocol (Kim *et al*, 2013). In 5C-Oct4GFP$^+$ cells, the expression of markers of pluripotency (*endo-Oct4*, *endo-Sox*, and *Nanog*) and markers of the naïve state (*endo-Klf4*, *Pecam1*, and *Tbx3*) was similar to those in R1 ESCs, while the expression of markers of the primed state (*Fgf5*, *Otx2*, and *FoxA2*) was similar to those in R1 ESCs-primed (Fig 5B). In addition, 5C-Oct4GFP$^+$ cells retained the XaXa state (Fig 5C), but seldomly contributed to chimeras (Fig 5D).

As summarized in Fig 5E, 5C-Oct4GFP$^+$ cells had characteristics of both naïve and primed PSCs, suggesting that 5C medium induces MEFs to a distinct pluripotent state (named 5C state), which may be between the naïve and primed states.

Two previously reported pluripotent states, epiblast-like state and formative state, are also important. To further compare the 5C-Oct4GFP$^+$ cells with epiblast-like cells (EpiLCs) and cells in formative state, previous reported datasets were downloaded and analyzed together with the current RNA-Seq (Data ref: Hayashi *et al*, 2011; Data ref: Buecker, 2014; Data ref: Bertone, 2014; Data ref: Kalkan *et al*, 2019). Based on these datasets and previous reports (Hayashi *et al*, 2011; Kalkan *et al*, 2017; Smith, 2017; Du *et al*, 2018; Takahashi *et al*, 2018), cells in 5C state are quite different from EpiLCs and cells in formative state. Generally, cells in 5C state seldomly contributed to chimera and had high expression of glycolysis and mesenchymal markers (Fig EV3A).

In addition, the gene expression in cells in these three states was normalized against those of ESCs, and the top 1,000 upregulated genes in each type of cells were selected. Less than 50% of the top 1,000 upregulated genes in 5C-Oct4GFP$^+$ cells were identified in EpiLCs or formative state cells (Fig EV3B–E). In addition, 5C-Oct4GFP$^+$ cells had a closer expression profile to primed PSCs than the other two types of cells. Approximately 70% of the top 1,000 upregulated genes in 5C-Oct4GFP$^+$ cells were also upregulated in R1 ESCs-primed. Furthermore, Gene Ontology (GO) studies suggested that the genes upregulated in only one type of cells were enriched with genes related to different fields (Fig EV3F). Genes

which were specifically upregulated in 5C-Oct4GFP$^+$ cells, EpiLCs, and cells in formative state were enriched with genes related to cell migration & extracellular matrix, apoptosis & ATP binding, and cell cycle & oxidation reduction, respectively (Fig EV3F).

5C-Oct4GFP$^+$ cells were quite heterogeneous, and the fluorescence intensities of REX1 and NANOG differed even in one particular colony (Appendix Fig S1B). Thus, the co-expression of naïve and primed markers in 5C-Oct4GFP$^+$ cells might be due to the fact that 5C-Oct4GFP$^+$ cells were a mixture of naïve and primed PSCs. To exclude this possibility, single-cell qPCR was performed with 5C-Oct4GFP$^+$ cells, R1 ESCs, and R1 ESCs-primed (Fig EV3G and H). *Dppa3*, *Esrrb*, *Fbxo15*, *Klf2/4/5*, *Nanog*, *Prdm14*, *Tbx3*, *Tfcp2l1*, and *Zfp42* were used as naive markers, while *Cer1*, *Fgf5*, *Foxa2*, *Lef1*, *Nodal*, *Sox1*, and *T* were used as primed markers. Approximately 55% of tested 5C-Oct4GFP$^+$ cells expressed naïve and primed markers at levels above average, while only approximately 12% R1 ESCs and approximately 2% R1 ESCs-primed had similar expression pattern (Fig EV3G). High expression of five naïve markers (*Esrrb*, *Klf2*, *Prdm14*, *Tbx3*, and *Tfcp2l1*) and three primed markers (*Cer1*, *Foxa2*, and *T*) were less frequently observed in 5C-Oct4GFP$^+$ cells than the other markers (Fig EV3H and Dataset EV3), which suggested these markers are better markers for naïve and primed states and further studies on these marker in these three states are required. Therefore, 5C-Oct4GFP$^+$ cells express several naïve and primed markers simultaneously at single-cell level.

**Naïve, primed, and new 5C states are interconvertible**

To verify whether 5C-Oct4GFP$^+$ cells could be converted into naïve or primed PSCs, 5C-Oct4GFP$^+$ colonies were cultured with naïve or primed medium (Dataset EV1). The two media converted 5C-Oct4GFP$^+$ cells to cells with typical morphologies of naïve and primed PSCs, and the obtained cells were stable for at least 20 additional passages (Fig 5F). RNA-Seq (GSE103765 and GSE103791) analysis suggested that the naïve and primed medium converted 5C-Oct4GFP$^+$ cells toward the naïve and primed states, respectively (Fig 5G), which was further confirmed by qPCR of the markers of different pluripotent states (Fig EV4A). In addition, primed medium converted 5C-Oct4GFP$^+$ cells to a XaXi state (Fig 5H), while naïve medium enabled 5C-Oct4GFP$^+$ cells to contribute to chimeras (Fig 5I and J). Embryoid body (EB) differentiation experiments suggested that the obtained cells had differentiation potential similar to that of naïve or primed PSCs (Fig EV4B and C). Therefore, the new pluripotent state is convertible to both naïve and primed states.

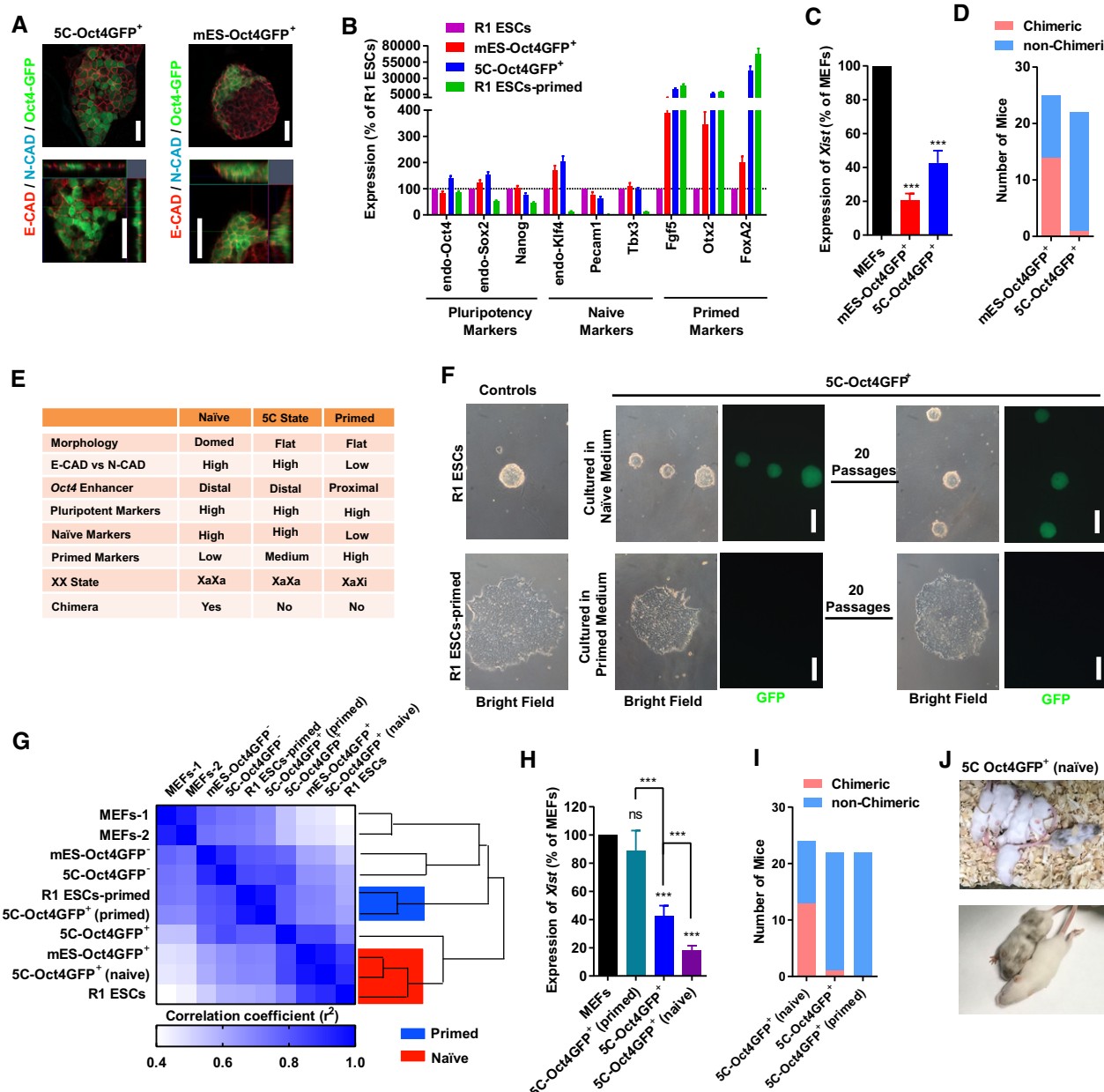

**Figure 5. 5C state is a distinct pluripotency state.**

A–E On day 15 during reprogramming, the expression of E-cadherin (E-CAD) and N-cadherin (N-CAD) was determined with immunofluorescence together with the morphology in Oct4GFP+ colonies (A, Scale bar, 25 μM). After sorting Oct4GFP+ cells out (5C-Oct4GFP+ and mES-Oct4GFP+ cells), the expression of markers of different pluripotent states (B) and *Xist* (C), and their abilities to form chimeras (D) were determined. The characteristics of 5C-Oct4GFP+ cells were summarized and compared to naïve and primed PSCs in (E).

F The isolated 5C-Oct4GFP+ cells were cultured in naïve or primed medium for 3 days, and the morphology changed toward R1 ESCs (naïve PSCs) and R1 ESCs-primed (primed PSCs), respectively. The obtained cells were maintained for additional 20 passages. The 5C-Oct4GFP+ cells cultured in primed medium lost GFP fluorescence. R1 ESCs-primed were primed PSCs generated from R1 ESCs. Scale bar, 100 μM.

G PSCs mentioned in (F) were used for RNA-Seq analysis. Similarity among gene expression profiles was determined.

H–J The isolated 5C-Oct4GFP+ cells were cultured in naïve or primed medium for 3 days to generate 5C-Oct4GFP+ (naïve) and 5C-Oct4GFP+ (primed) cells. The expression of *Xist* (H) and the abilities to form chimeras (I and J) were determined.

Data information: Experiments were independently repeated at least five times (*n* ≥ 5) except for sequencing and animal studies. Error bars represent standard deviations. ***$P < 0.001$. Additional statistical information is listed in Dataset EV7.

In addition, 3-day treatment with 5C medium converted R1 ESCs and R1 ESCs-primed toward the 5C state (Fig EV4D and E). 5C medium increased the expression of naïve markers in R1 ESCs-primed, while it increased the expression of primed markers in R1 ESCs. Therefore, either state of the three pluripotent states mentioned above could be converted into the other two.

## Highly efficient generation of naïve and primed PSCs

Because 5C-Oct4GFP$^+$ cells could be converted into both naïve and primed states, we then investigated reprogramming when the 5C medium was replaced with naïve or primed medium on day 10 (Fig EV5A).

To simplify the studies, generated colonies were classified into seven different types (Fig EV5B). Type A-Oct4GFP$^+$ and type A-Oct4GFP$^-$ colonies had relatively compact morphology and were conventional colonies induced with mES medium, while type B-Oct4GFP$^+$ and type B-Oct4GFP$^-$ colonies had loose morphology and were typical colonies induced with 5C medium. Type

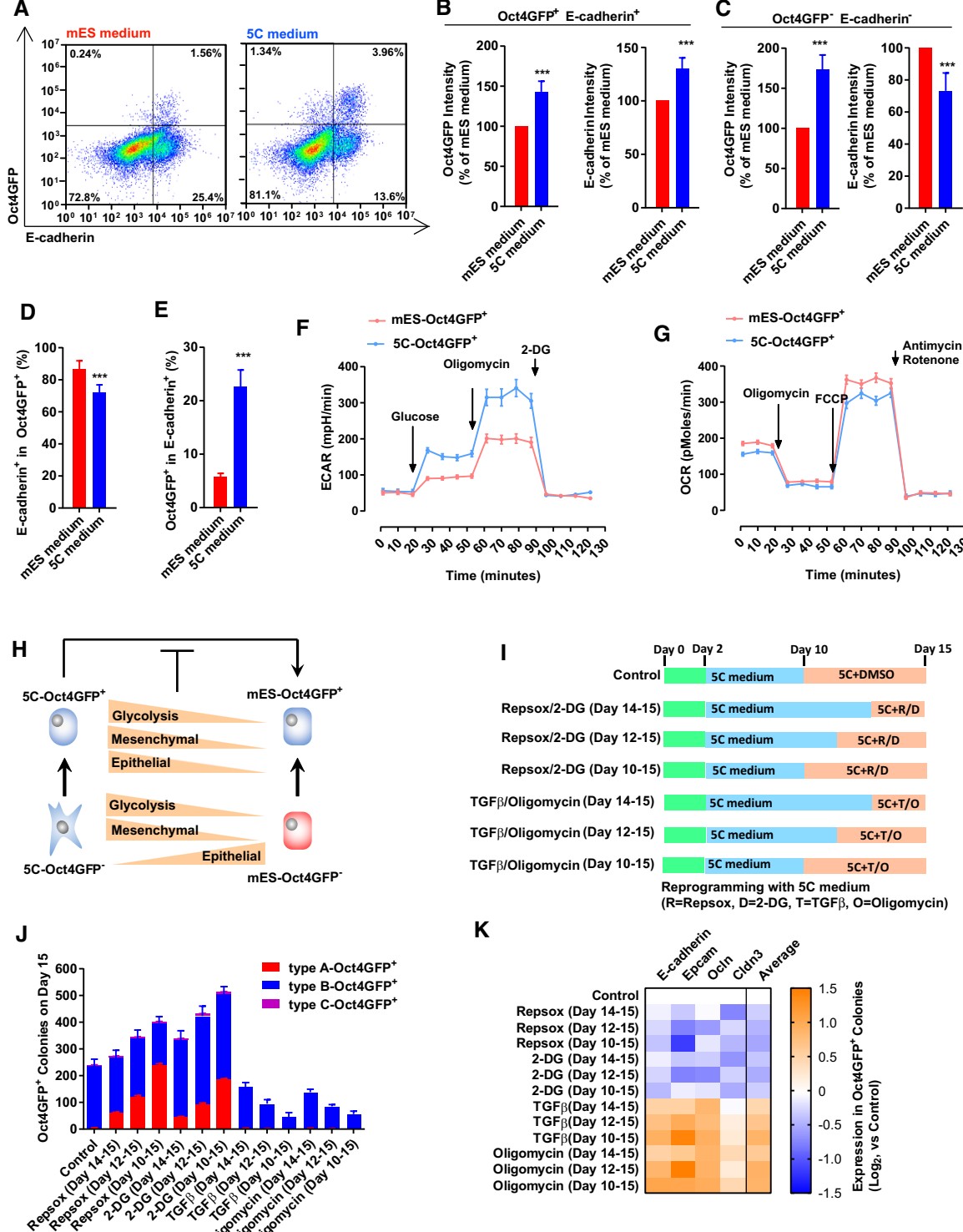

**Figure 6.**

Figure 6. EMT and OGS cooperate to inhibit reprogramming at the late stage.

A–E   The expression Oct4GFP and E-cadherin was determined with FACS on day 15 during reprogramming (A). Based on the FACS results, the expression of Oct4GFP and
       E-cadherin was summarized in Oct4GFP$^+$E-cadherin$^+$ cells (B) and in Oct4GFP$^-$E-cadherin$^-$ cells (C). The percentage of E-cadherin$^+$ cells in Oct4GFP$^+$ cells was
       listed in (D), while the percentage of Oct4GFP$^+$ cells in E-cadherin$^+$ cells was listed in (E).

F, G   Energy metabolism was analyzed in 5C-Oct4GFP$^+$ and mES-Oct4GFP$^+$ cells with Seahorse instrument.

H      5C-Oct4GFP$^-$ cells had higher expression of glycolysis and mesenchymal markers, but lower expression of epithelial markers than mES-Oct4GFP$^-$ cells. 5C-
       Oct4GFP$^+$ cells had higher expression of all three groups of markers than mES-Oct4GFP$^+$ cells.

I–K    EMT and OGS were modulated with TGFβ (TGFβ1/2/3, 1 ng/ml each), RepSox (1 μM), oligomycin (1 μM), and 2-DG (5 mM) at the late stage during reprogramming
       with 5C medium (I). The numbers of three types of Oct4GFP$^+$ colonies (J) and the expression of four epithelial markers in Oct4GFP$^+$ cells (K) were determined on
       day 15. "Average" in (K) suggested the average expression of four epithelia markers.

Data information: Experiments were independently repeated at least five times ($n \geq 5$). Error bars represent standard deviations. ***$P < 0.001$. Additional statistical
information is listed in Dataset EV7.

C-Oct4GFP$^+$ and type E-Oct4GFP$^-$ colonies had morphologies typical to those of naïve (compact and domed Oct4GFP$^+$) and primed PSCs (monolayer without Oct4GFP), respectively. Type D-Oct4GFP$^-$ colonies were even looser than type B and negative for Oct4GFP fluorescence.

The expression of markers of different pluripotent states was analyzed in the colonies (at least 12 individual colonies in each group). Type C-Oct4GFP$^+$ and type E-Oct4GFP$^-$ colonies had a similar expression to those of naïve and primed PSCs, respectively (Fig EV5C). Type A-Oct4GFP$^+$ colonies were closer to naïve PSCs, while type E-Oct4GFP$^-$ colonies were closer to primed PSCs. Type A-Oct4GFP$^-$, type B-Oct4GFP$^-$, and type D-Oct4GFP$^-$ colonies had low expression of all these markers. Therefore, the number of different types of colonies could be used to analyze the colonies generated with different protocols.

Although neither naïve nor primed medium induced a significant number of colonies when used alone, they generated many more colonies with typical naïve (type C-Oct4GFP$^+$) and primed (type E-Oct4GFP$^-$) morphologies when they were used to replace 5C medium on day 10 (Fig EV5D–F). When comparing the 5C-naïve and 5C groups, the percentage of Oct4GFP$^+$ cells increased from approximately 5% to approximately 40%, and the number of colonies with a typical naïve morphology (type C-Oct4GFP$^+$) increased from < 10 to more than 600. On the other hand, although primed medium decreased the percentage of Oct4GFP$^+$ cells by approximately 50%, it generated a significant amount of colonies with a typical primed morphology (type E-Oct4GFP$^-$).

The similarity between type C-Oct4GFP$^+$/E-Oct4GFP$^-$ colonies and naïve/primed PSCs was further demonstrated by the expression of Xist, and the ability to form chimeras (Fig EV5G and H). Thus, combinational usage of 5C and naïve/primed medium enables us to generate naïve/primed PSCs with high efficiency.

However, when naïve or primed medium was used to replace conventional mES medium on day 10 during reprogramming (Fig EV5I), similar effects were not observed. Naïve medium only slightly increased the percentage of Oct4GFP$^+$ cells from approximately 1.5% to approximately 2%, while primed medium did not induce colonies with typical primed morphology (Fig EV5J–L).

In addition, when early EMT or OGS was modulated during reprogramming by controlling the expression of Hif1α or using small-molecule compounds, consistent effects were observed. Facilitated EMT or OGS at the early stage of reprogramming generated more type B-Oct4GFP$^+$ colonies, while inhibited EMT or OGS resulted in more type A-Oct4GFP$^+$ colonies (Appendix Fig S6).

Thus, the generation of different types of Oct4GFP$^+$ colonies during reprogramming with the different medium was due to the early EMT and OGS.

The generation of iPSCs with 5C medium was compared with two previously defined medium iSF1 and iSF2 (also known as iCD1; Dataset EV1) (Chen et al, 2010, 2011). 5C and iSF1 medium induced Oct4GFP$^+$ colonies similarly, but preferred to generate type B- and type A-Oct4GFP$^+$ colonies, respectively (Appendix Fig S7A and B). In addition, when naïve medium was used to replace 5C medium on day 10 (5C-naïve), significantly more Oct4GFP$^+$ colonies were generated than iSF1 medium, and the generated colonies were type C-Oct4GFP$^+$ colonies (Appendix Fig S7B). However, iCD1 medium generated more Oct4GFP$^+$ colonies than 5C-naïve group, even when an additional 1:5 passage was performed on day 9 (Appendix Fig S7A and B). In addition, 5C, but not iSF1 and iCD1 medium, tended to induce early EMT as demonstrated by the expression of E-cadherin and N-cadherin on day 6 (Appendix Fig S7C). Thus, 5C medium induces reprogramming via a route different from that used by iSF1 or iCD1 medium.

Bone morphogenetic proteins (BMPs) promote reprogramming by inducing MET (Samavarchi-Tehrani et al, 2010), while also inhibit reprogramming by inducing pre-iPSCs (Chen et al, 2012). As reported previously, the beneficial effects of the BMP pathway are stronger than its inhibitory effects at low activation level, while the inhibitory effects are stronger at high activation level (Lin et al, 2018). Consistently, BMPs only promoted reprogramming with mES medium at a low concentration (Appendix Fig S7D). In addition, since 5C medium promoted reprogramming by inducing early EMT at least partially, the beneficial effects of the BMP pathway should be weaker during reprogramming with 5C medium than with mES medium. Thus, BMPs did not promote reprogramming with 5C medium at all concentrations (Appendix Fig S7D).

The beneficial effects of the BMP pathway, like inducing MET and activating pluripotency, can be achieved by Yamanaka factors and mES/5C medium at least partially, which makes BMPs redundant during reprogramming. Thus, activating the BMP pathway to a higher level only increased the inhibitory effects, but contributed little to the beneficial effects. However, inhibiting the BMP pathway with Noggin might block the beneficial effects which were contributed by both the BMP pathway and other factors. In addition, the beneficial effects of the BMP pathway are stronger than its inhibitory effects at a low activation level. Thus, impaired reprogramming was observed at all concentrations (Appendix Fig S7E), which is consistent with the previous report (Lin et al, 2018).

## OGS and EMT cooperated to inhibit further reprogramming at the late stage

Epithelial–mesenchymal transition at the late stage inhibited reprogramming (Liu *et al*, 2013). Thus, the cooperation between EMT and facilitated OGS might not be beneficial for the reprogramming at the late stage.

The cells on day 15 during reprogramming were tested for Oct4GFP and E-cadherin expression with FACS (Fig 6A). In Oct4GFP$^+$ E-cadherin$^+$ cells, the levels of the two proteins were higher in 5C-Oct4GFP$^+$ than in mES-Oct4GFP$^+$ cells (Fig 6B). In Oct4GFP$^-$ E-cadherin$^-$ cells, the level of Oct4GFP is higher, while the level of E-cadherin was lower in 5C-Oct4GFP$^+$ than in mES-Oct4GFP$^+$ cells (Fig 6C).

When the expression of mesenchymal and epithelial markers was determined in these four kinds of cells, consistent observations were obtained (Appendix Fig S8A and B). 5C-Oct4GFP$^+$ cells expressed higher levels of both mesenchymal and epithelial markers than mES-Oct4GFP$^+$ cells. 5C-Oct4GFP$^-$ cells expressed higher levels of mesenchymal markers, but lower levels of epithelial markers than mES-Oct4GFP$^-$ cells. Thus, we hypothesized that 5C medium prevents the downregulation of mesenchymal markers and forces cells to express a higher level of epithelial markers for compensation.

In addition, the percentages of Oct4GFP$^+$ E-cadherin$^+$ cells in Oct4GFP$^+$ and in E-cadherin$^+$ cells were determined. 5C medium had a lower ability to upregulate E-cadherin or induce MET in Oct4GFP$^+$ positive cells, and a higher ability to induce pluripotency in E-cadherin$^+$ cells than mES medium (Fig 6D and E). Thus, 5C medium may inhibit further reprogramming by preventing necessary MET, which also provides an explanation for the loose morphology of type B colonies induced with 5C medium.

5C-Oct4GFP$^+$ cells had higher glycolysis and lower oxidative phosphorylation than mES-Oct4GFP$^+$ cells (Fig 6F and G), which was confirmed with the expression of glycolysis and OXPHOS markers (Appendix Fig S8C and D, and Dataset EV4).

Thus, possibly because of the strong and positive feedback between EMT and OGS, 5C-Oct4GFP$^+$ cells had higher expression of mesenchymal, epithelial, and glycolysis markers than mES-Oct4GFP$^+$ cells (Fig 6H and Dataset EV4).

Epithelial–mesenchymal transition and OGS were then modulated during the late stage of reprogramming (Fig 6I). Inducing EMT or OGS at the late stage inhibited further reprogramming, while inhibiting these two events promoted reprogramming (Fig 6J). Such time-dependent effects were consistent with previous reports on EMT and OGS (Liu *et al*, 2013; Mathieu *et al*, 2014).

The higher expression of epithelial markers in 5C-Oct4GFP$^+$ cells might be due to the requirement to counteract the higher expression of mesenchymal markers or glycolysis markers. To confirm this hypothesis, TGFβ and RepSox were used for different amounts of time at the late stage during reprogramming (Fig 6I). Longer TGFβ treatment resulted in less Oct4GFP$^+$ colonies and higher expression of epithelial markers in the final Oct4GFP$^+$ colonies, while RepSox had the opposite effects (Fig 6J and K).

The abnormities in 5C-Oct4GFP$^+$ cells (higher expression of mesenchymal, epithelial, and glycolysis markers) made them unstable. We failed to culture 5C-Oct4GFP$^+$ colonies in 5C medium for more than two passages (Appendix Fig S9A and B). We then tried to culture 5C-Oct4GFP$^+$ colonies in 5C medium with 5% O$_2$ and TGFβ to maintain their abnormities. Then, the 5C-Oct4GFP$^+$ colonies could be maintained for five passages (1:2) with a significantly prolonged cell cycle (Appendix Fig S9A and B), suggesting that these abnormities are key characteristics of 5C-Oct4GFP$^+$ cells. However, culture conditions still required further optimization.

Facilitating OGS or inducing early EMT resulted in more Oct4GFP$^+$ colonies. However, the increased Oct4GFP$^+$ colonies were mainly in 5C state (type B-Oct4GFP$^+$ colonies; Fig EV5E). In addition, inhibiting OGS or inducing MET at the late stage converted type B-Oct4GFP$^+$ colonies to the type A-Oct4GFP$^+$ colonies which were relatively stable (Fig 6J). Besides, replacing 5C medium at the late stage also converted PSCs in 5C state to other states with higher stabilities (Fig EV5). Therefore, facilitated OGS and early EMT cooperate to promote pluripotency induction or the generation of PSCs in 5C state at the early stage, while OGS and EMT cooperate at the late stage to prevent further reprogramming from the new pluripotency state (5C state) to naïve state.

Both *Hif1α* and *Hif2α* had higher expression in 5C GFP$^+$ colonies than in mES GFP$^+$ colonies (Appendix Fig S10A). In addition, 5C medium had higher abilities to upregulate *Hif2α* than to upregulate *Hif1α* (Appendix Fig S10A). Similar observations were also identified on day 6 during reprogramming (Appendix Fig S10A). Besides, *Hif2α* had higher abilities to upregulate mesenchymal markers, while lower abilities to upregulate the five epigenetic factors (Appendix Fig S10B and C). Thus, *Hif1α* may be more important than *Hif2α* for the cooperation between OGS and EMT at the early stage, while *Hif2α* may be more important than *Hif1α* at the late stage. This hypothesis is supported by the previous report that HIF1α promotes reprogramming during the whole reprogramming process, while stabilization of HIF2α during late stages represses reprogramming (Mathieu *et al*, 2014).

## 5C-Oct4GFP+ cells are close to the epiblasts of E4.5 and E5.5 embryos

Since 5C-Oct4GFP$^+$ cells could be converted into both naïve and primed PSCs, and were considered to be in a distinct pluripotent state which locates between naïve and primed states (Fig 5), 5C-Oct4GFP$^+$ cells were further compared with naïve (R1 ESCs) and primed PSCs (R1 ESCs-primed). The OXPHOS in 5C-Oct4GFP$^+$ cells was lower than in naïve PSCs, but higher than in primed PSCs (Fig 7A). However, glycolysis in 5C-Oct4GFP$^+$ cells was higher than both naïve and primed PSCs (Fig 7B). Similar observations were confirmed with the expression of OXPHOS and glycolysis markers (Fig 7C and Dataset EV4). Similarly, although the expression of mesenchymal markers in 5C-Oct4GFP$^+$ cells was between those in the two PSCs, the expression of epithelial markers in 5C-Oct4GFP$^+$ cells was higher than those in naïve and primed PSCs (Fig 7C and Dataset EV4).

5C-Oct4GFP$^+$ cells had higher expression of epithelial and glycolysis markers than either naïve or primed PSCs (Fig 7C). Thus, although pluripotent cells in the 5C state have characteristics of both naïve and primed PSCs, the new state cannot be simply considered as an intermediate state. The RNA-Seq results of 5C-Oct4GFP$^+$ cells were compared with more than 200 types of mouse cells or tissues (Hutchins *et al*, 2017). The highest correlation was observed with ESCs (Dataset EV5), suggesting that the new 5C state may be a

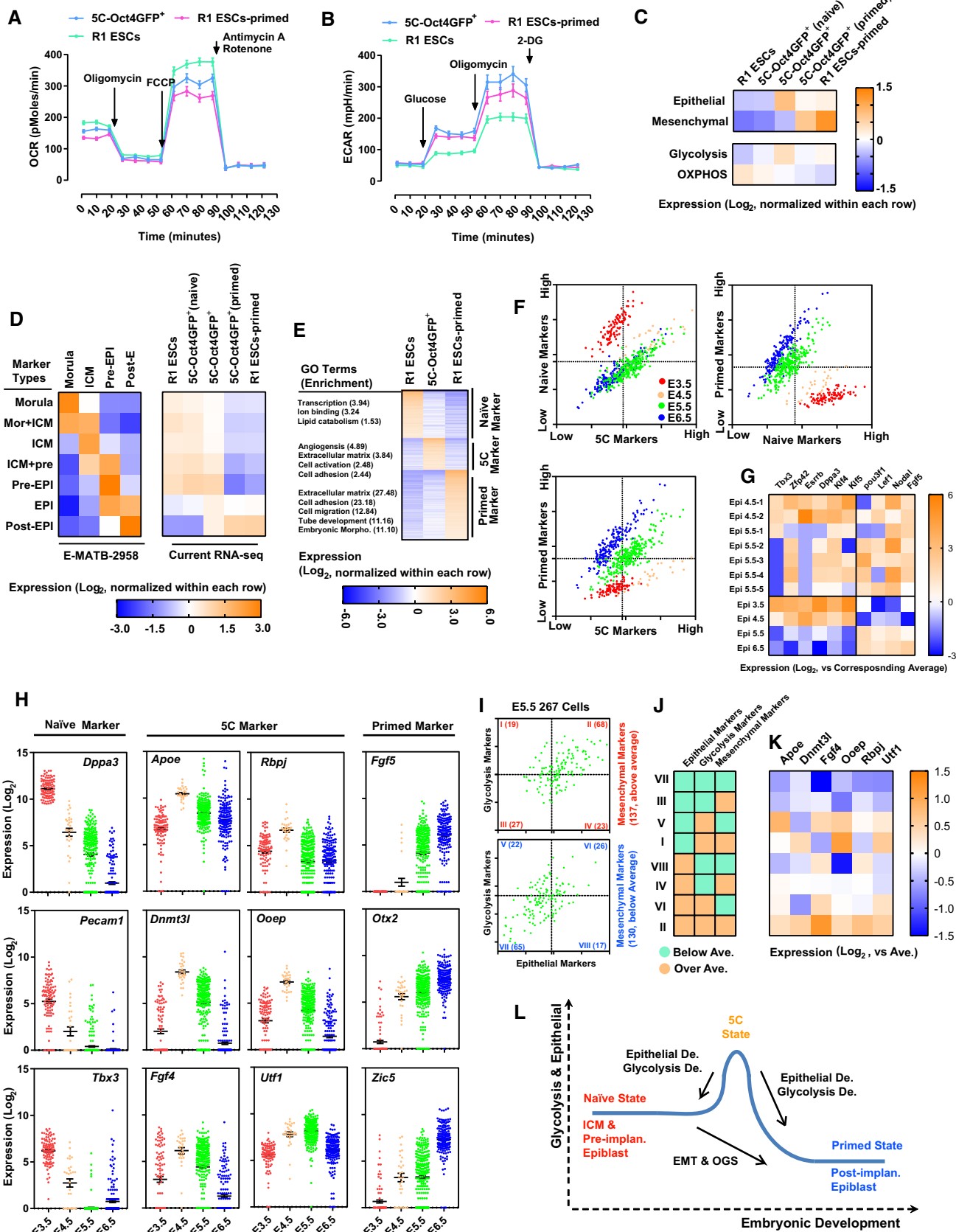

**Figure 7.**

◀

**Figure 7. 5C state is similar to the epiblast from E4.5/5.5 embryos.**

A, B Energy metabolism was analyzed with 5C-Oct4GFP⁺, R1 ESCs (naïve PSCs), and R1 ESCs-primed (primed PSCs) with Seahorse instrument.

C Expression of epithelial, mesenchymal, glycolysis, and oxidative phosphorylation (OXPHOS) markers (listed in Dataset EV4) was summarized in R1 ESCs, 5C-Oct4GFP⁺ (naïve), 5C-Oct4GFP⁺, 5C-Oct4GFP⁺ (primed), and R1 ESCs-primed, based on the current RNA-Seq. For each marker, Log₂ values of FKPM were normalized by subtracting the average of five groups. Log₂ values of FKPM of each group of markers were then averaged and plotted. 5C-Oct4GFP⁺ (naïve) and 5C-Oct4GFP⁺ (primed) were 5C-Oct4GFP⁺ cultured in naïve and primed medium, respectively, for 3 days.

D Genes with higher expression at different developmental stages were selected out from previously reported RNA-Seq (Data ref: Bertone, 2015). "Mor", "ICM", "pre-EPI", and "post-EPI" represented genes that have high expression only in morula, inner cell mass, pre-implantation epiblast, and postimplantation epiblast, respectively. "Mor + ICM", "ICM + pre", and "EPI" indicated genes that have high expression in both morula and inner cell mass, in both inner cell mass and pre-implantation epiblast, and in both pre-implantation and postimplantation epiblast, respectively. The results of previously reported RNA-Seq were summarized in the left panel, while the expression of these genes in current RNA-Seq was summarized in the right panel. Log₂ values of FKPM were presented similarly as in (C).

E Markers of 5C, naïve and primed states were selected based on the current RNA-Seq (Dataset EV4). The expression of these markers in R1 ESCs, 5C-Oct4GFP⁺, and R1 ESCs-primed was listed together with the GO enrichment of each group of markers. Log₂ values of FKPM were presented similarly as in (C).

F Markers of 5C, naïve, and primed states (Dataset EV4) were used to analyze expression profiles of epiblast during the early development of mouse embryos (Data ref: Mohammed *et al*, 2017). Single cell was plotted based on the average expression of each group of markers.

G The average expression of both naïve and primed markers was calculated based on GSE100597. The top two cells from E4.5 epiblast and top five cells from E5.5 epiblast were selected. The expression of several naïve markers (*Dppa3, Esrrb, Klf4, Klf5, Tbx3,* and *Zfp42*) and several mesenchymal markers (*Fgf5, Lef1, Nodal,* and *Pou3f1*) in these selected cells were listed (upper part). The average expression of these genes in E3.5, 4.5, 5.5, and 6.5 epiblast was also listed (lower part). These results were all normalized against the average expression of these genes in all tested cells.

H The expression of representative markers of 5C, naïve, and primed states was summarized in E3.5, 4.5, 5.5, and 6.5 epiblast.

I–K Cells from E5.5 epiblast were separated into eight groups (I–VIII) based on the average expression of epithelial, mesenchymal, and glycolysis markers (H). The average expression of three groups of markers was summarized in (J). The expression of six important markers of 5C state was summarized in these eight groups of cells (K)

L Schematic illustration of different pluripotent states and developmental stages.

Data information: Experiments were independently repeated at least five times (*n* ≥ 5) except sequencing. Error bars represent standard deviations. Additional statistical information is listed in Dataset EV7.

distinct pluripotency state rather than a state similar to other cells or tissues.

We then tried to determine the correlation between the 5C state and different stages during embryonic development (Data ref: Bertone, 2015). As indicated in Fig 7D, seven categories of genes with specific and higher expression at different developmental stages were analyzed. Consistent with the expression profile of ESCs reported previously (Boroviak *et al*, 2015), the current R1 ESCs and R1 ESCs-primed had the best correlation with pre- and postimplantation epiblasts, respectively (Fig 7D). However, 5C-Oct4GFP⁺ cells seemed to have expression profiles between pre- and postimplantation epiblasts. The genes with high expression in both pre- and postimplantation epiblasts also had high expression in 5C-Oct4GFP⁺ cells. Thus, 5C-Oct4GFP⁺ cells might represent the epiblast that lies between pre- and postimplantation epiblasts.

We then tried to confirm the correlation identified above. Markers for naïve, primed, and the new 5C state were selected based on the current RNA-Seq and subjected to GO analysis (Fig 7E and Dataset EV4). These markers were then used to analyze the single-cell RNA-Seq results during mouse embryonic development (Data ref: Mohammed *et al*, 2017; Mohammed *et al*, 2017). Epiblasts isolated from embryonic day (E) 4.5 and E5.5 embryos showed higher expression of 5C markers (Fig 7F). In addition, approximately 3% of E4.5 cells (1 in 29) and 25% of E5.5 cells (69 in 267) expressed both naïve and primed markers above average (Fig 7F). Besides, if we combined naïve and primed markers together, the cells with the highest expression of these markers were selected and they did express several key naïve and primed markers at high levels simultaneously (Fig 7G).

*Dppa3, Pecam1,* and *Tbx3* were three genes identified as naïve markers in current RNA-Seq and had higher expression in E3.5 epiblasts (Fig 7H). *Apoe, Dnmt3l, Fgf4, Rbpj, Ooep,* and *Utf1* were genes identified as 5C markers and had higher expression in E4.5

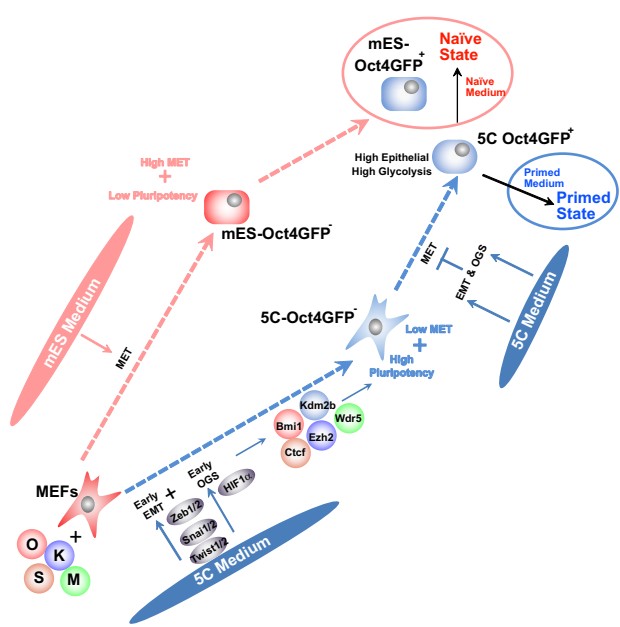

**Figure 8. Schematic illustration of current results.**

and E5.5 epiblasts (Fig 7H). *Fgf5, Otx2,* and *Zic5* were identified as primed markers and had higher expression in E6.5 epiblasts (Fig 7H).

The 267 cells isolated from E5.5 epiblasts were classified into eight groups based on the expression of epithelial, mesenchymal, and glycolysis markers (Fig 7I and J, and Dataset EV4). Cells were significantly enriched in Group II (approximately 26%, high expression of all three groups of markers) and Group VII (approximately

25%, low expression of all three groups of markers). When the expression levels of the six 5C markers listed in Fig 7H were analyzed, the highest and lowest expression levels were identified in Groups II and VII, respectively (Fig 7K). The abnormality of 5C-Oct4GFP$^+$ cells (higher expression of mesenchymal, epithelial, and glycolysis markers) was consistent with some cells from E5.5 epiblasts.

Therefore, the naïve, primed, and 5C states had the best correlation with pre-implantation epiblasts, postimplantation epiblasts, and epiblasts between these two stages, respectively (Fig 7L). The correlation between the 5C state and specific epiblasts was consistent with the low stability of the 5C state, which was possibly due to the abnormally higher expression of both epithelial and mesenchymal markers.

# Discussion

Since sequential EMT-MET has been extensively discussed during embryonic development and cancer development (Chaffer *et al*, 2007; Thiery *et al*, 2009), we hypothesize that early EMT poises the cells in a state that is beneficial for subsequent cell fate conversions (Liu *et al*, 2013; Li *et al*, 2014b). This hypothesis is supported by the critical roles of sequential EMT-MET during the transition from MEFs to neuron-like cells, the differentiation from ESCs to hepatocytes, and the conversion of human gastric epithelial cells to multipotent endodermal progenitors (Wang *et al*, 2016; He *et al*, 2017; Li *et al*, 2017).

The current study suggested that early EMT and facilitated OGS form a positive feedback loop, cooperate to upregulate several epigenetic factors including *Bmi1*, *Ctcf*, *Ezh2*, *Kdm2b*, and *Wdr5*, remove epigenetic barriers during pluripotency induction, and promote reprogramming at the early stage (Fig 8). The positive feedback loop between EMT and OGS is too strong and inhibits further reprogramming at the late stage by inducing glycolysis to a level that is too high for further reprogramming and by preventing necessary MET (Fig 8). However, since OGS and MET are highly connected with each other and with most singling pathways or transcriptional network, the current studies cannot exclude other possible mechanisms underlying the functions of 5C medium.

Although the cells in the new 5C state can be converted into naïve or primed PSCs, the 5C state is not simply an intermediate or averaged state of the naïve and primed states. Cells in the 5C state have higher expression of epithelial and glycolysis markers than naïve and primed PSCs. Such abnormalities of the 5C state do not suggest that it is an intermediate state. In addition, the cells in the new 5C state can be maintained with 5% O$_2$ and TGFβ and are similar to the epiblasts isolated from E4.5/E5.5 mouse embryo. The high reprogramming efficiency when combining different media (Fig EV5) also disagrees with the artificial hypothesis.

The number of cells increased during embryonic development from ICM to the pre-implantation and postimplantation phase. Because nutrition and oxygen can be supplied to the embryo via the placenta after implantation, the pre-implantation and early postimplantation epiblasts may have the lowest supply of oxygen and the highest degree of glycolysis (Fig 7) (Folmes *et al*, 2012), which is similar to the hyperglycolysis of the 5C state.

Gastrulation is the reorganization of the postimplantation epiblast into the three germ layers and involves significant cell rearrangement and EMT (Thiery *et al*, 2009). In the mouse, the blastocyst implants in the uterine wall on E4.5 and gastrulation begins on E6.5. Therefore, epiblasts isolated from E4.5/E5.5 mouse embryo are just before gastrulation. It is possible that the expression of mesenchymal markers was already increased, while the simultaneously increased expression of epithelial markers prevented the cells from entering EMT or the embryo from entering gastrulation. If this hypothesis is correct, pre-implantation epiblasts and cells in the 5C state should have high expression of epithelial markers.

Mesenchymal–epithelial transition is an early and beneficial step during reprogramming from MEFs to iPSCs. Introducing EMT, a reversed process of MET, before conventional MET surprisingly promoted reprogramming (Liu *et al*, 2013). We explained this observation by suggesting that sequential EMT-MET or early EMT poised the cells in a state that was beneficial for further cell fate conversion. After completing the current investigation, the beneficial roles of early EMT could be explained, at least partially. Early EMT induced OGS and cooperated with OGS to remove the epigenetic barriers during pluripotency induction. Such a mechanism is also suitable for explaining the beneficial roles of early EMT in other cell fate conversion processes.

The early EMT is considered to be on day 3 as indicated by the higher expression of mesenchymal markers (Fig EV2A and B) and larger cell migration (Fig EV2C) than MEFs. After the early EMT (day 6–15), MET did take place, but is slower and weaker than that induced by mES medium (Fig EV2A–C). Since the virus-containing medium was replaced with 5C or mES on day 2, most experiments were performed on day 6 rather than day 3 in order to provide sufficient time for the reprogramming system to function. In addition, the differences in EMT and OGS between reprogramming with mES and 5C medium seem to be larger on day 6 than on day 3.

H3K9 hypermethylation on pluripotency loci is a critical barrier during reprogramming and can be relieved by Vc. Since Vc is included in both mES medium and 5C medium, H3K9 hypermethylation cannot explain the different abilities of the two media to induce pre-iPSCs. The cooperation between early EMT and facilitated OGS activates additional epigenetic factors and promotes the conversion from pre-iPSCs (generated during reprogramming with mES medium) to iPSCs. *Bmi1*, *Ctcf*, *Ezh2*, *Kdm2b*, and *Wdr5* may help pre-iPSCs overcome additional barriers. Such observations extend our understanding of epigenetic changes during reprogramming.

# Materials and Methods

The information related to the materials, the assay kits, and the primers used in current studies is listed in Dataset EV6.

### Animal studies

All procedures related to animal studies were performed in accordance with the National Institutes of Health Guide for the Care and Use of Laboratory Animals (NIH Publication No. 80-23) and were approved by the Institutional Review Board in Guangzhou Institutes

of Biomedicine and Health. All efforts were made to minimize the number of animals used and their suffering.

Standard blastocyst injections were performed as described previously (Behringer *et al*, 2013). Briefly, chimeras were produced by injecting cells into blastocysts derived from ICR mice, followed by implantation into pseudopregnant ICR mice. Mice were normally housed in groups of four with access to food and water *ad libitum*. After implantation, mice were housed individually with access to food and water *ad libitum*. More than 20 mice were generated in each group, and chimeras were identified based on coat color. No specific randomization step or blinding was used for animal studies.

Mouse embryonic fibroblasts in the current study were generated by removing head and all internal organs from 13.5-day mouse embryos as described previously (Liu *et al*, 2013; Li *et al*, 2014a). MEFs were cultured in MEF medium for one passage before freezing in liquid nitrogen. MEFs were recovered from liquid nitrogen before use. All primary cultures were subjected to mycoplasma tests (MycoAlert™, Lonza) to ensure they were free of mycoplasma before use.

## Generation of iPSCs

Mouse embryonic fibroblasts were derived from 13.5-day mouse embryos carrying the Oct4GFP transgenic allele (Szabo *et al*, 2002). Retrovirus was packed by plat-E cells. MEFs within two passages were plated in 12-well plates ($1.5 \times 10^4$ cells/well) and then infected with retrovirus encoding *Oct4*, *Sox2*, *Klf4*, and *c-Myc* twice on day 0 and day 1, respectively. mES or 5C medium was used from day 2, and half-medium was replaced every day. The four Yamanaka factors were delivered simultaneously except when "OK + M + S" was used indicated. Reprogramming efficiency was evaluated by counting the number of Oct4GFP$^+$ colonies on days 11–15, AP$^+$ colonies on day 15, and detecting the percentage of Oct4GFP$^+$ cells by FACS on day 15.

The retrovirus system was also used to deliver exogenous expression or sh-RNA to control the expression of target genes. During reprogramming, the cDNA or sh-RNA containing retrovirus was used simultaneously with those contain cDNA of the four Yamanaka factors on days 0 and 1. The retrovirus system was also used to deliver exogenous expression or sh-RNA to MEFs on day 0, and qPCR and other experiments were performed on day 3.

## Cell culture

The medium used in the current studies is all listed in Dataset EV1 with final concentrations of different components and the necessary information for purchase. 5C medium includes DMEM/F12 (1:1), N2 supplement (1%), bFGF (20 ng/ml), β-mercaptoethanol (55 μM), Vc (55 μg/ml), and LIF (1,000 unit/ml). MEFs and plat-E cells were cultured in MEF medium.

R1 ESCs were cultured in naïve medium which including LIF and 2i on plates coated with 0.1% gelatin (Millipore) (Ying *et al*, 2008). R1 ESCs were passaged every 4 days with 0.25% trypsin-EDTA (Gibco) and plated into plates pre-coated with 0.1% gelatin for 2 h at 37°C.

Primed state of R1 ESCs was induced on MEF feeder cells with primed medium and maintained on FBS (Gibco) in primed medium

supplemented with 3 μM CHIR99021 (Selleck) and 2.5 μM IWR-1 (Sigma), instead of bFGF (Peprotech) and activin A (R&D Systems) (Kim *et al*, 2013). Primed state stem cells were passaged every 3 days with 1.5 mg/ml collagenase IV (Gibco), triturated into small clumps, and plated into plates pre-coated with FBS, instead of gelatin.

## qPCR

Total RNA was extracted with TRIzol (Invitrogen), and 2 μg RNA was used for reverse transcription. Transcript levels of genes were determined on the CFX96 Real-Time System (Bio-Rad) with SsoAdvanced SYBR Green Supermix (Bio-Rad). The primers are listed in Dataset EV6.

## Immunostaining

Cells were cultured on cover glasses in 24-well cell culture plates, fixed with 4% PFA at room temperature for 10 min, and washed thrice with PBS. Rabbit anti-E-cadherin (Cell Signaling Technology, 3195, 1:200), mouse anti-N-cadherin (Invitrogen, 333900, 1:200), rabbit anti-Nanog (Novus Biologicals, NB100-58842, 1:300), and mouse anti-Rex1 (made in-house, 1:500) were added as the primary antibodies prepared with 0.3% Triton X-100 and 10% goat serum in PBS at 4°C overnight. Then, cells were washed thrice with PBS. Goat anti-rabbit Alexa 647 (Abcam, ab150079, 1:500) and goat anti-mouse Alexa 568 (Invitrogen, A11004, 1:500) were added as the secondary antibodies prepared with PBS at room temperature for 1 h. Then, cells were washed thrice with PBS. Nuclei were stained with DAPI prepared in PBS at room temperature for 5 min. Image collecting was conducted with Zeiss LSM 800.

## Flow cytometry analysis

Cells were trypsinized to single cells with 0.25% trypsin-EDTA, fixed with 4% PFA at room temperature for 10 min, and permeabilized with 0.1% Triton X-100 at 4°C for 2 min. Then, cells were washed with 1% FBS and incubated with rabbit anti-E-cadherin as the primary antibody prepared with 1% FBS in PBS at 4°C for 2 h. Then, cells were washed with PBS and incubated with goat anti-rabbit Alexa 647 prepared with PBS at room temperature for 1 h. Then, cells were washed with PBS and resuspended in PBS. Flow cytometry analysis was performed with Accuri C6 (BD Biosciences).

## Embryoid body formation

For naïve PSCs, cells were treated with 0.25% trypsin-EDTA and diluted to $1 \times 10^5$ cells/ml with differentiation medium comprising of DMEM and 15% FBS. Then, the cells were dropped onto the lid of 150-mm cell culture plate containing PBS and incubated at 37°C for 3 days. For primed state, cells were triturated into small clumps with 1.5 mg/ml collagenase IV and cultured with differentiation medium in low-attachment cell culture plate at 37°C for 2 days. For 5C iPSCs, single GFP$^+$ colonies were picked and cultured with differentiation medium in low-attachment cell culture plate at 37°C for 3 days. Then, the embryoid bodies were transferred into a 100-mm cell culture plate and incubated at 37°C for 3 days.

## ChIP-qPCR

Chromatin from cells was fragmented to a size range of 200–500 bases with a Sonicator. Solubilized chromatin was immune-precipitated with antibody against H3K4me2 (Abcam, ab7766, 1:1,000), H3K4me3 (Abcam, ab8580, 1:1,000), H3K9me2 (Abcam, ab1220, 1:1,000), H3K9me3 (Abcam, ab8898, 1:1,000), H3K27me3 (Millipore, 17-622, 1:2,000), and H3K36me3 (Abcam, ab9050, 1:1,000). Antibody–chromatin complexes were pulled down with protein A/G (Invitrogen), washed, and then eluted. After cross-link reversal and proteinase K treatment, immune-precipitated DNA was extracted with phenol–chloroform, ethanol precipitated, and treated with RNase. ChIP DNA was quantified using PicoGreen. For ChIP-qPCR, primer sequences are listed in Dataset EV6. qPCR was performed on the CFX96 Real-Time System (Bio-Rad) with SsoAdvanced SYBR Green Supermix (Bio-Rad).

## AP staining and TUNEL staining

Cells were fixed with 4% PFA at room temperature for 2 min and washed twice with TBST. Freshly prepared AP staining solution (1 ml AP buffer + 6.6 μl NBT + 3.3 μl BCIP) was added, and plates were incubated in the dark at room temperature for 5 min and then washed with PBS. Apoptotic cell death was analyzed by the TUNEL assay using the *in situ* Cell Death Detection Kit, TMR red (Roche).

## Cellular energy metabolism

The cellular energy metabolism was assessed using the Seahorse XF24 extracellular flux analyzer (Seahorse Bioscience) according to the instruction of instruments, with the simultaneous measurement of the OCR and the ECAR as the indicator of mitochondrial respiration and glycolytic conversion of glucose to lactate, respectively. For mitochondrial respiration, cells were seeded for 12 h in Seahorse XF24 cell culture plates and then the medium was replaced by XF assay medium (Seahorse Co.) supplemented with 25 mM glucose. Three injections were sequentially performed: 1 μM oligomycin (an ATP synthase inhibitor) at 30 min, 1 μM FCCP (a classic uncoupler of oxidative phosphorylation) at 60 min, and 1 μM rotenone (a complex I inhibitor) and 1 μM antimycin A (a complex III inhibitor) were simultaneously injected at 90 min. The following OCR parameters were analyzed and calculated. For glycolytic conversion of glucose to lactate, medium was replaced by XF assay medium without glucose or pyruvate. The assay workflow was as follows: 10 mM glucose was injected at 30 min, 1 μM oligomycin at 60 min, and 50 mM 2-DG (a glucose analog) at 90 min. The following ECAR parameters were analyzed and calculated. Increase in ECAR after adding glucose indicated the level of glycolysis of the cells, while a decrease in OCR after adding oligomycin indicated the level of ATP production or OXPHOS of the cells.

## RNA-Seq

RNA was extracted from cells with TRIzol. Illumina mRNA-Seq libraries were prepared using the TruSeq RNA Sample Preparation Kit v2 (Illumina). Then, the samples were sequenced on an Illumina MiSeq instrument with the MiSeq Reagent Kit (Illumina). Obtained RNA-Seq reads were processed by RSEM (RNA-Seq by expectation–maximization) to estimate transcript abundances. Reads were aligned to a synthetic transcriptome, and the number of reads associated with a given transcript was used to estimate that transcript's abundance in TPM (transcripts per million).

## Migration

The migration abilities of cells were determined with live-cell imaging with a Cell Observer (Zeiss, Germany). During the tracing, bright-field pictures were captured every hour. The individual cells were identified in each image with the location of the geometric gravity centers. The distance between the locations of the particular cell (hour $n$ and hour $n + 1$) was considered as the migration ability of the cell. Cell tracing was achieved with self-developed software and help from manual correction. If untraceable, the closest cell on hour $n + 1$ was used for calculation.

## HIF1α activity

Reporter for HIF1α activity was prepared in a Lentivirus system (Zhou *et al*, 2011). The reporter is YFP driven by minimal HIF-binding sites and minimal human thymidine kinase promoter. The reporter was delivered into MEFs 1 day before (day −1) the delivery of Yamanaka factor during reprogramming. The YFP expression was determined on days 0, 3, and 6 with FACS.

## Single-cell qPCR

Specific target amplification and reverse transcription for sc-qPCR was performed using Single-Cell Sequence-Specific Amplification Kit (Vazyme, P621-01) following the manufacturer's instructions. Briefly, the Assay Pool containing the various sets of primers as described in Dataset EV6 were prepared by mixing forward and reverse primers and diluted with nuclease-free ddH2O to give a final concentration of 100 nM of each primer. Reverse transcription and pre-amplification were performed in a final volume of 5 μl. Single cells were prepared with flow cytometry (BD FACS, Aria IIU). The pre-amplified complementary DNA (cDNA) was diluted 1:5 with nuclease-free ddH2O and stored at −20°C.

BioMark Dynamic Array 48.48 GE IFC chip (Fluidigm, BMK-M-48.48) was used according to the manufacturer's instructions. Briefly, assay mixture contained 2.5 μl 10 μM combined primers and 2.5 μl 2× Assay Loading Reagent (Fluidigm, PN 100-7611) was prepared. Sample pre-mix was made by combining 2.5 μl 2× SsoFast EvaGreen Supermix with low ROX (Bio-Rad, PN 172-5211), 0.25 μl 20× GE Sample Loading Reagent (Fluidigm, PN 100–7610), and 2.25 μl pre-amplified cDNA for each of the 48 sample inlets. IFC controller (Fluidigm, MX) was used to prime 48.48 dynamic arrays IFC Chip with control line fluid. Five microliter of each assay and sample mix was then transferred into the appropriate inlets of the primed chip and loaded with the IFC controller. After loading, the chip was placed in the Biomark instrument for Fluidigm screening at 95°C for 1 min followed by 30 cycles at 96°C for 5 s and 60°C for 20 s. The data were analyzed with Real-Time PCR Analysis Software in the Biomark instrument (Fluidigm).

## Statistical information

Experiments were conducted at least five independent biological repeats with the exception of RNA-Seq analysis. Each group in

chimera studies included at least 20 mice. Data were analyzed and compared by Student's *t*-tests or one-way ANOVA with Dunnett's test as a *post hoc* test or two-way ANOVA with Bonferroni's test as a *post hoc* test with GraphPad Prism 7.0. Error bars represent standard deviations, and "*n*" represents the number of independent experiments. "*", "**", and "***" represent significant differences ($P < 0.05$), ($P < 0.01$), and ($P < 0.001$) from indicated control groups, respectively. Detailed statistical information is provided in Dataset EV7.

## Data availability

RNA-Seq data: Gene Expression Omnibus GSE103765 (https://www.ncbi.nlm.nih.gov/geo/query/acc.cgi?acc=GSE103765). RNA-Seq data: Gene Expression Omnibus GSE103791 (https://www.ncbi.nlm.nih.gov/geo/query/acc.cgi?acc=GSE103791).

**Expanded View** for this article is available online.

## Acknowledgements

This work was supported by the Strategic Priority Research Program of Chinese Academy of Sciences (Grant No. XDA16010305), the Key Research Program of Frontier Sciences of Chinese Academy of Sciences (Grant No. QYZDB-SSW-SMC031), the National Natural Science Foundation of China (Grant No. U1601228, 31671475, 31421004, 31900699, and 81702445), the Key and Development Program of Guangzhou Regenerative Medicine and Health Guangdong Laboratory (Grant No. 2018GZR110104008), and the Science and Technology Planning Project of Guangdong Province (Grant No. 2017B030314056).

## Author contributions

HZ and DP conceived and supervised the study and wrote the manuscript. HZ designed the experiments and analyzed the data. HS, FM, and XL characterized the new 5C state and performed bioinformatics analysis. XY, MZ, FW, and TY performed experiments on reprogramming and pre-iPSCs conversion, and determined the contribution of EMT and OGS; LNL, YL, and JC determined the cooperation between EMT and OGS. CL, JH, MH, QX, QL, and LLL repeated several experiments to confirm the results.

## Conflict of interest

The authors declare that they have no conflict of interest.

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
