## [Review Process File · The EMBO Journal]

Metabolic switch and epithelial-mesenchymal transition cooperate to regulate pluripotency

Hao Sun, Xiao Yang, Lining Liang, Mengdan Zhang, Yuan Li, Jinlong Chen, Fuhui Wang, Tingting Yang, Fei Meng, Xiaowei Lai, Changpeng Li, Jingcai He, Meiai He, Qiaoran Xu, Qian Li, Lilong Lin, Duanqing Pei, Hui Zheng

Review timeline:	Submission date:	16th Jul 2019
	Editorial Decision:	9th Sep 2019
	Revision received:	2nd Dec 2019
	Editorial Decision:	13th Jan 2020
	Revision received:	21st Jan 2020
	Accepted:	23rd Jan 2020

Editor: Daniel Klimmeck

Transaction Report:

1st Editorial Decision

9th Sep 2019

Thank you for the submission of your manuscript (EMBOJ-2019-102961) to The EMBO Journal. Please accept my apologies for the unusual delay in processing your manuscript at this time of the year due to protracted referee input. Your manuscript has been sent to three reviewers, and we have received reports from all of them, which I enclose below.

As you will see, the referees acknowledge the potential interest and novelty of your results, although they also express a number of major issues - both on the experimental/analysis and manuscript structure sides - that will have to be addressed before they can support publication of your manuscript in The EMBO Journal. In more detail, referee #2 states that characterization of the intermediate 5C state is too preliminary and requires more refined experimentation to be conclusive (ref#2, pts 3,4, 6-1ff; see also ref#1, pt.3). Further, this referee finds that your claims regarding a feedback loop between EMT and glycolytic control factor is not sufficiently supported (ref#2, pt.5). Referee #3 agrees in that the HIF1a-OGS factor interplay needs to be better explored and asks you to consolidate causalities between transcriptional, metabolic and epigenetic regulation (ref#3, pts. 2,3). In addition, the reviewers raise a number of issues related to methods annotation, data representation, statistics and appropriate citation of literature references as well as clarity of the overall manuscript that would need to be conclusively addressed to achieve the level of robustness and clarity needed for The EMBO Journal.

I judge the comments of the referees to be generally reasonable and given their overall interest, we are happy to invite you to revise your manuscript experimentally to address the referees' comments.

REFeree REPORTS:

Referee #1:

In this manuscript the authors investigate the effects of EMT and the switch from oxidative phosphorylation to glycolysis on reprogramming. In the course of this, they identify a pluripotent state that is neither naïve nor primed and which they surmise is an intermediary state of pluripotency.

The identification of an intermediary state between naïve and primed pluripotency is of interest, particularly given that such a state is known to exist, at least transiently both in vivo at the time of implantation and in vitro in the form of epiblast-like cells (EpiLCs, Hayashi et al.). Given this, it is a surprise that the authors do not discuss such cells, particularly as others have speculated on whether such cells can be cultured (Smith, Development). A problem with the manuscript is in the conflation of the analysis of this intermediate state with the EMT/OGS study which makes both parts (which may be able to stand alone) less prominent than they perhaps optimally could be.

In addition, the manuscript is difficult to follow as there are numerous places where insufficient description of the data is provided either in the figure, in the legends and in the text to comprehend easily what individual experiments consist of. I would recommend that the manuscript is significantly re-written to eliminate ambiguities in all the figures and ensure that the results adequately describe the experiments performed.

SPECIFIC COMMENTS

Major

1 the abstract is unclear. In particular, the difference in meaning between sentences 1 and 3 is obscure, making sentence 2 also unclear. This has a major impact on the comprehensibility of the rest of the abstract.

2 The authors provide a considerable body of data in the study. However, much of it is inadequately described in the figures, legends and text for its value to be fully apparent. I am going to restrict my comments here to Figure 1, but the same general shortcomings affect all the data figures. In panels C and D, there is no explanation of the abbreviations on the y-axes in the legends, so it is unclear what is being measured here; this detail belongs on the legend not in the M+M. In G and K, there is no description of the many abbreviations used in the diagrams, nor is there sufficient description of the methods used to be helpful, either in the legend or in the M+M. One might guess that in G Hif1a is being overexpressed, but this is not stated anywhere. J+N: y-axes should say Oct4-GFP colonies.

3 The composition of 5C medium and the concentration of components is given in Dataset EV1. As the composition of 5C medium is an original finding for pluripotent cell culture and will therefore be of interest to others, it would be appropriate to also list the composition in the materials and methods section. This section already contains detailed compositions for naïve and primed culture, neither of which are original to this study.

Minor

4 GFP refers to Oct4GFP throughout and should therefore be relabelled in figures and legends and in the text throughout the manuscript.

5 On page 5 the text should probably read '5C-GFP- cells had closer gene expression profiles to iPSCs or ESCs than mESC-GFP- cells'

6 also on page 5 the text should possibly read 'the low expression and high H3K9 methylation of core pluripotency loci in pre-iPSCs are critical, etc'

7 Extended data figure 1 B should also show GFP- staining.

8 if q-RTPCR is being done in extended data Figure 2 B and C, this should be stated clearly in the legend.

Referee #2:

In this study, the authors describe iPSC generation in the presence of the chemically defined "5C medium" and examine the underlying cause for its beneficial effect on reprogramming. The authors demonstrate that the 5C medium promotes a metabolic switch from oxidative phosphorylation to glycolysis (OGS) and consequently the induction of HIF1a, which in turn induced expression of epigenetic modifiers that are known to facilitate reprogramming. In addition, the authors propose that EMT and OGS form a positive feedback loop that can facilitate early stages of reprogramming. When analysing the pluripotent state of the reprogrammed 5C cells, the authors detect expression of

both primed and naive pluripotency marker genes. Comparing gene expression in the 5C state to expression in early embryonic development, the authors observed the greatest similarity to the peri-implantation epiblast, leading to the conclusion that the 5C medium confers the reprogrammed cell a novel pluripotent state distinct from naive or primed pluripotency.

This study adds on previous publications describing the importance of the metabolic state and EMT for iPSC generation (Folmes et al. 2011; Liu et al. 2013), by linking a switch in glucose metabolism to EMT and epigenetic reprogramming, which is of importance for a broad spectrum of developmental processes. The postulation of a novel 5C pluripotent state is interesting but premature and lacks key experimental evidence to substantiate the hypothesis.

Major comments

1) Chen and colleagues published in 2012 (PMID: 23202127) serum free conditions for iPSC generation. How does the iPSC generation efficiency in iSF1/2 medium (Chen et al. 2012) compare to the efficiency obtained with 5C medium?

2) It has been shown before that BMPs within the FBS in the conventional mES medium prevents pre-iPSCs from activating the endogenous pluripotency network and consequently interfere with iPSC generation (Chen et al. 2012). Does BMP signalling interfere with OGS or early EMT induction? Does the addition of Noggin alone or in combination with OGS induction further promote iPSC generation in mES or 5C medium?

3) As suggested by the authors early EMT and/or OGS should promote, while late EMT/OGS should inhibit iPSC generation (Fig.2J and Fig.5D). However, a different effect is observed in Fig.2I of the manuscript. Here, pre-iPSCs were generated in the presence of mES-Vc or mES medium and subsequent treated with TGFbeta or HIF1a to induce EMT/OGS. In contrast, to the authors model induction of late OGS/EMT promotes in this experiment iPSC generation (Fig.2I).

4) The authors suggest that 5C medium promotes early EMT in reprogramming MEFs through HIF1a (Fig.1F; Fig.3E). However, judging by A.) the migratory potential (Fig.3B), B.) the mesenchymal marker gene expression e.g. Snail1/2 (Fig.3C), and C.)the higher mesenchymal identity of MEFs(Fig. EV6B) it seems that MEF process more mesenchymal characteristics than reprogramming cells cultured in 5C or mES medium . Hence, rather than inducing EMT, the 5C medium preserves the mesenchymal MEF character initially better than conventional mEC medium.

5) The expression of Snail1/2 and Twist1/2 is significantly lower in 5C medium than in MEFs (Fig.3C). How can these EMT factors form a positive feedback loop with HIF1a/OGS when their expression is decreasing?

6) The authors suggest that MEF reprogramming in 5C medium results in a new pluripotent state (5C state), which combines attributes of primed and naive pluripotency. While this hypothesis is interesting, further in-depth characterisation of the 5C cells is essential to support the authors conclusion.

6-1.) Analysis of heterogeneity in the GFP positive 5C cell population. The GFP half-life is about 24h (Corish et al. 1999), and hence it is possible that cells undergoing differentiation remain GFP positive for a certain time. To consolidate their finding of a novel pluripotent state, it is absolutely essential to show co-expression of naive and primed pluripotency marker genes on a single cell level (Immunostaining or single cell qRT-PCR).

6-2.) While it is possible that some marker genes for primed and naive pluripotency are co-expressed, it is unlikely that the entire primed and naive pluripotency gene regulatory network (GRN) is simultaneously active in one cell as parts of these GRNs opposing each other (e.g. PRDM14, Yamaji et al 2013). It is important to see more comprehensively, which naive and primed marker 5C cells co-express (e.g. Dppa3, Essrb, Prdm14, Zfp42, Pou3f1, Lef1, Sox3 etc.)

6-3.) In the past Austin Smith's lab has investigated a transitory pluripotent state termed "Formative pluripotency", which exhibits characteristic to the peri-implantation epiblast and thereby bridges naive and classical primed pluripotency (Kalkan and Smith, 2014 Smith 2017, Kalkan et al. 2019). As the authors also detect the highest similarity between 5C cells and the E4.5/E5.5 epiblast, it

would be interesting to see how similar or different the 5C state is from the "Formative pluripotent" state suggested by the Smith lab.

6-4.) When reanalysing the single cell RNA-seq data from Mohammed et al. 2017, can the authors identify a subpopulation of cells at E4.5 or E5.5 that co-expresses naive and primed pluripotency marker genes characteristic for the 5C state?

minor points

1) The authors show that the 5C medium promotes HIF1a expression and suggest that HIF1a in turn regulates expression of epigenetic modifiers such as Wdr5 directly. In the absence of HIF1a ChIP data, indirect effects e.g. through HIF1a-induced TGFb expression (Fig. EV3A and Mingyuan et al. 2018) cannot be excluded and should be mentioned.

2) The authors have generated a remarkable amount of data. However, the description of the experimental design and obtained results is often insufficient. Describing the conducted experiments in the figure legends in greater detail and introducing key technical features of the work e.g. the GOF18ΔPE transgene and AP-staining as means to discriminate between pre-iPSCs and iPSC would be beneficial for the reader.

3) Fig. 1H/1L: The measured value (likely ECAR [mpH/min]) should appear on the Y-axis

4) Fig. EV1B: In addition to the merge, it would be good to show the Oct4/Nanog/Rex1 staining/channel separately. This could help to assess potential heterogeneity within the 5C colonies. Scale bars are missing in this panel.

5) Fig.2H upper part: As GFP expression is the readout in this experiment the figure should state "iPSC colonies" instead of "pre-iPSC colonies"

6) Fig.3F: It should be stated in the figure legend where MEFs were transfected with OSKM simultaneously or sequentially.

7) Fig.4H: It is unclear which significance level belongs to which comparison

8) Fig.6F. It is unclear what is plotted here on the X- and Y-axis. A more detailed description in the figure legend would be beneficial

9) page 18: When the authors describe the higher expression of Hif1a/2a in 5C colonies it should be referred to Fig.EV6E instead of EV5E

10) page 22: ... Gastrulation is the segregation of the "post-implantation epiblast" into three germ layers ...

Referee #3:

Sun et al. investigate mechanisms by which a chemically defined medium (5C) enhances the efficiency of reprogramming of MEFs to iPSCs. In particular, they explore roles of a metabolic shift from oxidative phosphorylation to glycolysis (OGS) and epithelial-mesenchymal transition (EMT), both of which they find to be increased in 5C as compared to traditional reprogramming culture media. While OGS and EMT were previously reported to promote reprogramming, the authors now explore their interconnection and provide evidence that OGS and EMT reinforce each other and act together in inducing several epigenetic factors that accelerate reprogramming. The authors further follow up on their previous findings that EMT inhibits late stages of reprogramming and demonstrate that OGS contributes to this phenomenon. As a consequence, high EMT and OGS trap cells in an intermediate stage that appears to have similarity to embryonic epiblast cells. The interplay between metabolism, epigenetics and transcriptional control in shaping reprogramming efficiency and stem cell identity is a timely and interesting subject. The authors present a large amount of data that carefully document how modulating OGS and EMT alone and together affects reprogramming efficiency. The results give novel and interesting insights that help

connect previously isolated findings concerning these processes. The correlation between the 5C state and epiblast cells seems rather speculative, but the progress in iPSC generation reported in the present work should be of great interest to the stem cell community.

Major points:

1. The relevance of OGS as compared to other transcriptional activities of HIF1a is not clear. To address this, the authors should increase / inhibit glycolysis in the context of HIF1a over-expression or knockdown and determine the effects on reprogramming in this context. Along these lines, to strengthen the experiments concerning the postulated feedback loop between HIF1a/OGS and EMT, HIF1a and EMT transcription factors should be manipulated together and in the context of increased/inhibited glycolysis, and reprogramming efficiency tested. In particular, is there any input from metabolism beyond the transcriptional activities of HIF1a and the EMT transcription factors into expression of the identified epigenetic factors (5F)?

2. In Figure 2, the authors identify 5 epigenetic factors with diverse activities / functions that are upregulated in 5C medium and contribute to reprogramming efficiency. I understand that the present work focuses on improving reprogramming protocols, but it would be helpful to have at least some indication that the observed changes in 5F levels result in epigenetic changes and altered expression of pluripotency factors.

3. The authors conclude that 5F function downstream of EMT and OGS - does this mean that they function independently of metabolic state, if they are expressed? This is of particular interest in the context of the present study, because metabolite levels can influence the abundance of epigenetic marks directly, independently of altered transcription of the enzymes that add / erase these marks.

Minor points:

1. p. 12 Figure S4F should be S3F

Additional suggestions:

The readability of the manuscript could be improved by discussing the data in more detail and separating data presentation from interpretation more clearly. Two examples to make this point:

- p. 6, Figure 1C-D: '5C medium induced a larger metabolic switch from oxidative phosphorylation to glycolysis than mES medium'. Figure 1C shows a clear increase in ECAR, whereas Figure 1D shows but a minor decrease in OCR, suggesting that glycolysis is increased whereas oxidative phosphorylation is largely unperturbed. The measured parameters should be explicitly referred to and the obtained results discussed separately.
- p. 12, Figure 3D: 'These methods modulated OGS similarly to HiF1a or sh-HiF1a but induced EMT to a lesser extent.' The figure shows a quantification of cell migration as a proxy for EMT, which should be made clear in the text.

1st Revision - authors' response

2nd Dec 2019

Response to Reviews' Comments

Editorial Summary

As you will see, the referees acknowledge the potential interest and novelty of your results, although they also express a number of major issues - both on the experimental/analysis and manuscript structure sides - that will have to be addressed before they can support publication of your manuscript in The EMBO Journal. In more detail, referee #2 states that characterization of the intermediate 5C state is too preliminary and requires more refined experimentation to be conclusive (ref#2, pts 3,4, 6-1ff; see also ref#1, pt.3). Further, this referee finds that your claims regarding a

feedback loop between EMT and glycolytic control factor is not sufficiently supported (ref#2, pt.5). Referee #3 agrees in that the HIF1a-OGS factor interplay needs to be better explored and asks you to consolidate causalities between transcriptional, metabolic and epigenetic regulation (ref#3, pts. 2,3). In addition, the reviewers raise a number of issues related to methods annotation, data representation, statistics and appropriate citation of literature references as well as clarity of the overall manuscript that would need to be conclusively addressed to achieve the level of robustness and clarity needed for The EMBO Journal.

Response: Thanks for The EMBO Journal for giving us the opportunity to address the concerns raised by the reviewers and submit the revised manuscript.

In the revised manuscript, we discussed more about the differences among the current 5C state, the epiblast-like state, and the formative state (Figure EV3A-C). In general, the current 5C state has distinct characteristics from the epiblast-like state and the formative state. For example, the cells in the new 5C state seldom contribute to chimeras and have high expression of glycolysis and mesenchymal markers. In addition, we further identified the differences of the current 5C state from the other pluripotent states (epiblast-like state, formative state, naïve, and primed state) with single-cell qPCR and provided additional data in Figure EV3D-E. The detailed responses to comments (ref#2, pts 3,4, 6-1; see also ref#1, pt.3) were provided below.

In the revised manuscript, additional data was provided to demonstrate the feedback loop between EMT and glycolytic regulators. For the comment (ref#2, pt.5), it is because the experiments involved the conversion from pre-iPSCs and iPSCs were not described clearly enough in the original manuscript. Such conversion is not equal or highly related to the ability of the feedback loop to inhibit reprogramming at the late stage. Additional discussion was provided below and in the text.

To address the comment (ref#3, pts. 2,3), we determined the contribution of the metabolic state to the epigenetic regulation in the revised manuscript. In addition, the interaction among EMT modulation, *Hif1a* expression regulation, and metabolic control during reprogramming was further explored (Appendix Figure S5).

In addition, we found that some concerns raised by the reviewers were because of the low clarity of the original manuscript, we have re-written the manuscript to fix the problem.

The point-to-point response was provided below.

Referee #1:

In this manuscript the authors investigate the effects of EMT and the switch from oxidative phosphorylation to glycolysis on reprogramming. In the course of this, they identify a pluripotent state that is neither naïve nor primed and which they surmise is an intermediary state of pluripotency.

The identification of an intermediary state between naïve and primed pluripotency is of interest, particularly given that such a state is known to exist, at least transiently both in vivo at the time of implantation and in vitro in the form of epiblast-like cells (EpiLCs, Hayashi et al.). Given this, it is a surprise that the authors do not discuss such cells, particularly as others have speculated on whether such cells can be cultured (Smith, Development). A problem with the manuscript is in the conflation of the analysis of this intermediate state with the EMT/OGS study which makes both parts (which may be able to stand alone) less prominent than they perhaps optimally could be.

Response: Thanks for the reviewer's comment. The cells in 5C state are generated during reprogramming from MEFs. The cells in epiblast-like and formative states are generated from ESCs or naïve PSCs. Thus, we only compared 5C state with naïve/primed state, but not with epiblast-like or formative state in the original manuscript.

In the revised manuscript, we discussed more about the differences among the current 5C state, the epiblast-like state and the formative state (Figure EV3A-C). We first listed the summarized characteristics of the three states in Figure EV3A. For example, the cells in new 5C state seldom contribute to chimeras and have high expression of glycolysis and mesenchymal markers. Then, by comparing the gene expression in cells in these three states to those in corresponding control ESCs (based on the current RNA-Seq and previous reported datasets), top-1000 upregulated genes in these three states (relative to ESCs) were selected. As indicated in Figure EV3B-E, the three groups of top-1000 upregulated genes were not overlapped with each other too much. In addition, 5C cells had a closer expression profile to primed PSCs than the other two types of cells (Figure EV3C&E). Both the microarray and RNA-Seq datasets from cells at the epiblast-like state and the formative state were used, and similar results were obtained. In addition, the genes which were only upregulated in one of the three states were enriched with genes related to distinct fields (Figure EV3F). Therefore,

in general, the current 5C state is different from the epiblast-like state and the formative state.

Additional discussion was provided in page 16 paragraph 4 to page 17 paragraph 2.

As suggested by the reviewer and previous reports, the formative state is difficult to maintain. However, Austin Smith group did not exclude the possibility to culture cells in formative state. As quoted directly from (Kalkan et al, 2017), "It will be of future interest to dissect in detail the molecular dynamics and drivers of transition in this defined and simple system and also to determine whether the formative phase may be suspended as a stem cell state in culture, as achieved for naïve ES cells and primed EpiSCs.". In addition, Austin Smith group captured the formative state for human PSCs (Rostovskaya et al, 2019). Since the current 5C state is distinct from the formative state, and cells in 5C state are generated during reprogramming instead of differentiation, it is reasonable for us to maintain it for several passages. The cells in 5C state can be maintained for 5 passages (1:2), but the cell cycle was significantly prolonged (Appendix Figure S9). Thus culture condition requires further optimization. Additional discussion was provided in page 24 paragraph 3.

In addition, the manuscript is difficult to follow as there are numerous places where insufficient description of the data is provided either in the figure, in the legends and in the text to comprehend easily what individual experiments consist of. I would recommend that the manuscript is significantly re-written to eliminate ambiguities in all the figures and ensure that the results adequately describe the experiments performed.

Response: We are sorry for the insufficient clarity. We have re-written the manuscript to fix the problem. Especially, we have re-organized the figures, provided more information in legends and in material and methods, and proofread the whole manuscript.

SPECIFIC COMMENTS

Major

1 the abstract is unclear. In particular, the difference in meaning between sentences 1 and 3 is obscure, making sentence 2 also unclear. This has a major impact on the comprehensibility of the rest of the abstract.

Response: Thanks for the reviewer's suggestion. We have modified the first three sentences in the abstract to the following "The metabolic switch from oxidative phosphorylation to glycolysis (OGS)

and epithelial-mesenchymal transition (EMT) promote reprogramming at the early stage. However, their connections have not been elucidated. Here, when a chemically-defined medium was used to induce early EMT during reprogramming, a facilitated OGS was also observed at the same time."

2 The authors provide a considerable body of data in the study. However, much of it is inadequately described in the figures, legends and text for its value to be fully apparent. I am going to restrict my comments here to Figure 1, but the same general shortcomings affect all the data figures. In panels C and D, there is no explanation of the abbreviations on the y-axes in the legends, so it is unclear what is being measured here; this detail belongs on the legend not in the M+M. In G and K, there is no description of the many abbreviations used in the diagrams, nor is there sufficient description of the methods used to be helpful, either in the legend or in the M+M. One might guess that in G Hif1a is being overexpressed, but this is not stated anywhere. J+N: y-axes should say Oct4-GFP colonies.

Response: Thanks for the reviewer's suggestion. We have proofread the manuscript and made necessary corrections. In the revised manuscript, we have described all the abbreviations and methods in the legends.

For example, we have discussed ECAR and OCR in the text (page 6 paragraph 4), in the legend and in the Materials and Methods (page 36 paragraph 2). In addition, we have changed "GFP" to "Oct4GFP" in the figures and text.

3 The composition of 5C medium and the concentration of components is given in Dataset EV1. As the composition of 5C medium is an original finding for pluripotent cell culture and will therefore be of interest to others, it would be appropriate to also list the composition in the materials and methods section. This section already contains detailed compositions for naïve and primed culture, neither of which are original to this study.

Response: Thanks for the reviewer's suggestion. We provided the detailed compositions for 5C medium both in Introduction (page 4 paragraph 3) and Materials and Methods section (page 32 paragraph 4) in revised manuscript.

Minor

4 GFP refers to Oct4GFP throughout and should therefore be relabeled in figures and legends and in the text throughout the manuscript.

Response: Thanks for the reviewer's suggestion. We have changed "GFP" to "Oct4GFP" in the figures and text.

5 On page 5 the text should probably read '5C-GFP- cells had closer gene expression profiles to iPSCs or ESCs than mESC-GFP- cells'

Response: We are sorry to make the mistake, we have corrected it.

6 also on page 5 the text should possibly read 'the low expression and high H3K9 methylation of core pluripotency loci in pre-iPSCs are critical, etc'

Response: We are sorry to make the mistake, we have corrected it.

7 Extended data figure 1 B should also show GFP- staining.

Response: Extended Data figure 1B (original) is the immunofluorescence for Oct4GFP, NANOG, and REX1. Oct4GFP- colonies have no detectable expression of Oct4GFP, NANOG, and REX1. Thus we only described in the legend but did not provide the image in revised manuscript in **Appendix Figure S1B**. In addition, we provided the images in separated channels in revised manuscript in **Appendix Figure S1B**.

8 if q-RTPCR is being done in extended data Figure 2 B and C, this should be stated clearly in the legend.

Response: We are sorry for this mistake. qPCR was performed in the extended figure 2B&C (original) or **Figure 3B&C** (revised). We have stated clearly in the legend and also fixed similar problems.

Referee #2:

In this study, the authors describe iPSC generation in the presence of the chemically defined "5C medium" and examine the underlying cause for its beneficial effect on reprogramming. The authors demonstrate that the 5C medium promotes a metabolic switch from oxidative phosphorylation to

glycolysis (OGS) and consequently the induction of HIF1a, which in turn induced expression of epigenetic modifiers that are known to facilitate reprogramming. In addition, the authors propose that EMT and OGS form a positive feedback loop that can facilitate early stages of reprogramming. When analysing the pluripotent state of the reprogrammed 5C cells, the authors detect expression of both primed and naive pluripotency marker genes. Comparing gene expression in the 5C state to expression in early embryonic development, the authors observed the greatest similarity to the peri-implantation epiblast, leading to the conclusion that the 5C medium confers the reprogrammed cell a novel pluripotent state distinct from naive or primed pluripotency.

This study adds on previous publications describing the importance of the metabolic state and EMT for iPSC generation (Folmes et al. 2011; Liu et al. 2013), by linking a switch in glucose metabolism to EMT and epigenetic reprogramming, which is of importance for a broad spectrum of developmental processes. The postulation of a novel 5C pluripotent state is interesting but premature and lacks key experimental evidence to substantiate the hypothesis.

Major comments

1) Chen and colleagues published in 2012 (PMID: 23202127) serum free conditions for iPSC generation. How does the iPSC generation efficiency in iSF1/2 medium (Chen et al. 2012) compare to the efficiency obtained with 5C medium?

Response: As suggested by the reviewer, we have provided the reprogramming efficiency with different serum-free medium including iSF1 and iSF2 (iCD1) in revised manuscript.

As indicated in Appendix Figure S7A-B, 5C medium and iSF1 medium generated approximately 250 Oct4GFP⁺ colonies on day 15. If we use the naïve medium to replace 5C medium on day 10 (5C-naïve), approximately 900 Oct4GFP⁺ colonies were induced. Because the iCD1 medium contains many more beneficial factors of reprogramming (Dataset EV1), it induced approximately 1200 Oct4GFP⁺ colonies on day 15. Since cells reprogrammed with iCD1 medium became confluent quicker than with other media, an additional 1:5 passage was performed on day 9.

iSF1 medium prefers to generate type A-Oct4GFP⁺ colonies, while 5C medium prefers to generate type B-Oct4GFP⁺ colonies. 5C-naïve protocol and iCD1 generates type C-Oct4GFP⁺ colonies. In addition, reprogramming with 5C medium, but not iSF1 and iCD1 medium, tended to induce early EMT as indicated by the expression of *E-cadherin* and *N-cadherin* on day 6 (Appendix

Figure S7C). Thus, 5C medium induces reprogramming via a route different from that used by iSF1 or iCD1 medium. Additional discussion was provided in page 21 paragraph 3.

2) It has been shown before that BMPs within the FBS in the conventional mES medium prevents pre-iPSCs from activating the endogenous pluripotency network and consequently interfere with iPSC generation (Chen et al. 2012). Does BMP signalling interfere with OGS or early EMT induction? Does the addition of Noggin alone or in combination with OGS induction further promote iPSC generation in mES or 5C medium?

Response: Bone morphogenetic proteins (BMPs) promote reprogramming by inducing MET (Samavarchi-Tehrani et al, 2010), while also inhibit reprogramming by inducing pre-iPSCs (Chen et al, 2012). As reported previously, the beneficial effects of the BMP pathway are stronger than its inhibitory effects at a low activation level, while inhibitory effects are stronger at high activation level (Lin et al, 2018). Consistently, BMPs only promoted reprogramming with mES medium at low concentration (Appendix Figure S7D). In addition, since 5C medium promoted reprogramming by inducing early EMT at least partially, the beneficial effects of the BMP pathway should be weaker during reprogramming with 5C medium than with mES medium. Thus, BMPs did not promote reprogramming with 5C medium at all concentrations (Appendix Figure S7D).

The beneficial effects of the BMP pathway, like inducing MET and activating pluripotency, can be achieved by Yamanaka factors and mES/5C medium at least partially, which makes BMPs redundant during reprogramming. Thus activating the BMP pathway to a higher level only increased the inhibitory effects, but contributed little to the beneficial effects. However, inhibiting the BMP pathway with Noggin might block the beneficial effects which were contributed by both the BMP pathway and other factors. In addition, the beneficial effects of the BMP pathway are stronger than its inhibitory effects at a low activation level. Thus, impaired reprogramming was observed at all concentrations (Appendix Figure S7E), which is consistent with previous report (Lin et al, 2018). Additional discussion was provided in page 21 paragraph 4 to page 22 paragraph 2.

3) As suggested by the authors early EMT and/or OGS should promote, while late EMT/OGS should inhibit iPSC generation (Fig.2J and Fig.5D). However, a different effect is observed in Fig.2I of the manuscript. Here, pre-iPSCs were generated in the presence of mES-Vc or mES medium and

subsequent treated with TGFbeta or HIF1a to induce EMT/OGS. In contrast, to the authors model induction of late OGS/EMT promotes in this experiment iPSC generation (Fig.2I).

Response: pre-iPSCs have been identified as AP+Oct4GFP- colonies during reprogramming. pre-iPSCs have typical iPSCs morphology but low expression of Oct4 and other pluripotency markers (Chen et al, 2012). Although pre-iPSCs were counted on day 15 during reprogramming, they appear as early as day 5 (Mikkelsen et al, 2008). Thus, the generation of pre-iPSCs and conversion from pre-iPSCs to iPSCs are not late events during reprogramming, they take place during the whole reprogramming process.

mES medium fails to help pre-iPSCs overcome certain barriers during reprogramming, while 5C medium can. Thus, 5C medium generated less pre-iPSCs or AP+Oct4GFP- colonies during reprogramming than mES medium (Figure 2A-B). In addition, we demonstrated a pathway from 5C medium, to early and facilitated OGS, to the upregulation of several key epigenetic factors, and to the decrease in pre-iPSCs (Figure 1-3, EV1). This pathway does not function during reprogramming with mES medium. Then, we tried to determine whether these modulating OGS/EMT and key epigenetic factors can convert the pre-iPSCs generated during reprogramming with mES medium, to iPSCs. The results in original Figure 2I and new Figure 3D-E confirmed that inducing EMT/OGS or overexpressing key epigenetic factors converted these pre-iPSCs to iPSCs, which provided another evidences to support that 5C medium reduces pre-iPSCs by inducing EMT and OGS. In addition, these methods failed to help the pre-iPSCs generated with 5C medium. Therefore, the pre-iPSCs generated with the two media are different in their response to EMT and OGS. Additional discussion was provided in page 11 paragraph 3.

In addition, since pre-iPSCs have typical colony morphology, they should have completed MET but not pluripotency induction. 5C medium induces sequential EMT-MET or early EMT, and does not induce many pre-iPSCs colonies during reprogramming (approximately 10% of all colonies). Thus, the OGS and EMT do not have enough pre-iPSCs to inhibit.

Most of the inhibitory effects of OGS and EMT at the late stage are that they impair the convention of type B-Oct4GFP+ colonies to type A- or type C-Oct4GFP+ colonies. 5C medium induced quicker and larger upregulation of pluripotency markers during reprogramming than mES medium (Figure EV1B). In addition, 5C medium had a low ability to induce MET in cells that already express core pluripotency genes (Figure 6A-E). In summary, the inhibitory effects of OGS

and EMT at the late stage prevent the MET but not the conversion from pre-iPSCs to iPSCs. We used "OGS and EMT cooperated to inhibit further reprogramming at the late stage" in text in page 22 paragraph 3 to page 23.

4) The authors suggest that 5C medium promotes early EMT in reprogramming MEFs through HIF1a (Fig.1F; Fig.3E). However, judging by A.) the migratory potential (Fig.3B), B.) the mesenchymal marker gene expression e.g. Snail1/2 (Fig.3C), and C.)the higher mesenchymal identity of MEFs(Fig. EV6B) it seems that MEF process more mesenchymal characteristics than reprogramming cells cultured in 5C or mES medium . Hence, rather than inducing EMT, the 5C medium preserves the mesenchymal MEF character initially better than conventional mEC medium.

Response: 5C medium induces sequential EMT-MET or early EMT during reprogramming. The early EMT is considered to be on day 3 as indicated by the higher expression of mesenchymal markers (Figure EV2A-B) and larger cell migration (Figure EV2C) than MEFs. After the early EMT (day 6-15), MET did take place, but is slower and weaker than that induced by mES medium (Figure EV2A-C).

Original Figure 3B (revised Figure 4B) and Figure 3C (revise Figure 4A) were performed on day 6, when MET already begins during reprogramming with 5C medium. Thus it is reasonable to find lower migration and expression of mesenchymal markers than MEFs. We have edited the legend to make them clearer.

Original Figure EV6B (revised Appendix Figure S8B) was performed on day 15 during reprogramming. 5C-Oct4GFP⁻ cells had significant MET, but lower than that in mES-Oct4GFP⁻ cells. Oct4GFP⁺ cells were different. 5C-Oct4GFP⁺ cells expressed higher levels of both mesenchymal and epithelial markers than mES-Oct4GFP⁺ cells, which is why 5C-Oct4GFP⁺ colonies had type B morphology while mES-Oct4GFP⁻ cells had type A morphology (Figure EV5). Such observations also explain why OGS and EMT inhibited further reprogramming at the late stage. We have edited the legend to make them clearer.

In addition, since the virus-containing medium was replaced with 5C or mES on day 2 (Materials and Methods), most experiments were performed on day 6 rather than day 3 in order to provide sufficient time for the reprogramming system to function. Most experiments were performed on day 6 rather than day 3 also because: 1) the differences in EMT and OGS between

reprogramming with mES and 5C medium seems to be larger on day 6 than on day 3; 2) cell proliferated a lot during day 3-6, which made the results more stable; 3) no colonies formation and Oct4GFP⁺ was observed on day 6, which suggested a better time point to study early event.

As suggested by the reviewer, the 5C medium's functions on day 6 can be described as the preservation of mesenchymal character. However, such effects on day 6 are consequences of the early EMT on day 3. Therefore we still used early EMT in the revised manuscript.

5) The expression of Snail1/2 and Twist1/2 is significantly lower in 5C medium than in MEFs (Fig.3C). How can these EMT factors form a positive feedback loop with HIF1 α /OGS when their expression is decreasing?

Response: The early EMT is considered to be on day 3 as indicated by the higher expression of mesenchymal markers (Figure EV2A-B) and larger cell migration (Figure EV2C) than MEFs. After the early EMT (on day 6-15), MET did take place, but is slower and weaker than that induced by mES medium (Figure EV2A-C). Original Figure 3C (new Figure 4A) was performed on day 6.

On day 3 during reprogramming with 5C medium, the expression of EMT factors and Hif1 α was higher than those in MEFs and cells in mES medium group. Thus, EMT factors and Hif1 α can form a positive feedback loop.

On day 6 or later during reprogramming with 5C medium, because the Yamanaka factors, like Sox2 and Klf4, induce MET (Liu et al, 2013), the early EMT switch to late MET. However, the MET with 5C medium is slower and weaker than that with mES medium (Figure EV2A-B). Although the expression of EMT factors was lower than those in MEFs, it was still higher than that in cells in mES medium group. The lower expression does not mean no expression, the positive feedback loop still exists between EMT factors and Hif1 α , but at a lower level. The positive feedback loop slowed the MET process of cells and prevented the further reprogramming from 5C state to naïve state.

6) The authors suggest that MEF reprogramming in 5C medium results in a new pluripotent state (5C state), which combines attributes of primed and naive pluripotency. While this hypothesis is interesting, further in-depth characterisation of the 5C cells is essential to support the authors conclusion.

6-1.) Analysis of heterogeneity in the GFP positive 5C cell population. The GFP half-life is about 24h (Corish et al. 1999), and hence it is possible that cells undergoing differentiation remain GFP positive for a certain time. To consolidate their finding of a novel pluripotent state, it is absolutely essential to show co-expression of naïve and primed pluripotency marker genes on a single cell level (Immunostaining or single cell qRT-PCR).

Response: Thanks for the reviewer's suggestion. We provided the immunofluorescence images in separated channels in the revised manuscript in **Appendix Figure S1B**. 5C-Oct4GFP⁺ cells were quite heterogeneous, the fluorescence intensities of REX1 and NANOG differed even in one particular colony (**Appendix Figure S1B**). Thus, the co-expression of naïve and primed markers in 5C-Oct4GFP⁺ cells might be due to the fact that 5C-Oct4GFP⁺ cells were a mixture of naïve and primed PSCs. To exclude this possibility, single-cell qPCR was performed with 5C-Oct4GFP⁺ cells, R1 ESCs, and R1 ESCs-primed (**Figure EV3G-H and Dataset EV3**).

Dppa3, Esrrb, Fbxo15, Klf2/4/5, Nanog, Prdm14, Tbx3, Tfcp2l1, and Zfp42 were used as naïve markers, while *Cer1, Fgf5, Foxa2, Lef1, Nodal, Sox1, and T* were used as primed markers. Approximately 55% of tested 5C-Oct4GFP⁺ cells expressed naïve and primed markers at levels above average, while only approximately 12% R1 ESCs and approximately 2% R1 ESCs-primed had similar expression pattern (**Figure EV3G**). High expression of five naïve markers (*Esrrb, Klf2, Prdm14, Tbx3, and Tfcp2l1*) and three primed markers (*Cer1, Foxa2, and T*) were less frequently observed in 5C-Oct4GFP⁺ cells than the other markers (**Figure EV3H and Dataset EV3**), which suggested further studies on these marker in these three states were required. Therefore, 5C-Oct4GFP⁺ cells express several naïve and primed markers simultaneously at single-cell level (**Page 18 Paragraph 1**).

In addition, according to the single cell RNA-Seq results during mouse embryonic development (Data ref: Gene Expression Omnibus GSE100597, 2017), approximately 3% of E4.5 cells (1 in 29) and 25% of E5.5 cells (69 in 267) expressed both naïve and primed markers above average (**Figure 7F**). In addition, if we combined naïve and primed markers together, the cells with the highest expression of these markers did express several key naïve and primed markers at high levels simultaneously (**Figure 7G**).

6-2.) While it is possible that some marker genes for primed and naïve pluripotency are co-

expressed, it is unlikely that the entire primed and naive pluripotency gene regulatory network (GRN) is simultaneously active in one cell as parts of these GRNs opposing each other (e.g. PRDM14, Yamaji et al 2013). It is important to see more comprehensively , which naive and primed marker 5C cells co-express (e.g. Dppa3, Essrb, Prdm14, Zfp42, Pou3f1, Lef1, Sox3 etc.)

Response: Thanks for the reviewer's suggestion. As indicated by in the single cell qPCR resulted provided in the revised manuscript (**Figure EV3G-H, Dataset EV3**), not all primed or naïve markers can be detected in 5C-Oct4GFP⁺ cells at high levels. High expression of five naïve markers (*Esrrb*, *Klf2*, *Prdm14*, *Tbx3*, and *Tfcp2l1*) and three primed markers (*Cer1*, *Foxa2*, and *T*) were less frequently observed in 5C-Oct4GFP⁺ cells than the other markers (**Figure EV3H and Dataset EV3**), which suggested further studies on these marker in these three states were required.

Only approximately 37.5%, 15.8%, 24.2%, 35.8%, and 29.2% 5C-Oct4GFP⁺ cells expressed *Esrrb*, *Klf2*, *Prdm14*, *Tbx3*, and *Tfcp2l1* at levels above the average in R1 ESCs cells. Approximately 33.3%, 22.5%, and 14.2% 5C-Oct4GFP⁺ cells expressed *Cer1*, *Foxa2*, and *T* at levels above the average in R1 ESCs-primed (**Dataset EV3**). These markers are better markers for naïve and primed states I (**Page 18 Paragraph 1**).

6-3.) In the past Austin Smith's lab has investigated a transitory pluripotent state termed "Formative pluripotency", which exhibits characteristic to the peri-implantation epiblast and thereby bridges naïve and classical primed pluripotency (Kalkan and Smith, 2014 Smith 2017, Kalkan et al. 2019). As the authors also detect the highest similarity between 5C cells and the E4.5/E5.5 epiblast, it would be interesting to see how similar or different the 5C state is from the "Formative pluripotent" state suggested by the Smith lab.

Response: Thanks for the reviewer's suggestion. In the revised manuscript, we discussed more about the differences among the current 5C state, the epiblast-like state and the formative state (**Figure EV3A-C**). We first listed the characteristics of the three states in (**Figure EV3A**). For example, the cells in new 5C state seldom contribute to chimeras and have high expression of glycolysis and mesenchymal markers. Then, by comparing the gene expression in cells of these three states to those in corresponding control ESCs (the current RNA-Seq and previous reported datasets), one array and one RNA-Seq datasets were used for the epiblast-like state and the formative state, top-1000 upregulated genes in these three states (compared to ESCs) were selected. As indicated in **Figure**

EV3B, the three groups of top-1000 upregulated genes were not overlapped with each other too much. In addition, the genes which were only upregulated in one of the three states were enriched with genes related to distinct fields (**Figure EV3C**). Therefore, in general, the current 5C state has distinct characteristics from the epiblast-like state and the formative state.

6-4.) When reanalysing the single cell RNA-seq data from Mohammed et al. 2017, can the authors identify a subpopulation of cells at E4.5 or E5.5 that co-expresses naive and primed pluripotency marker genes characteristic for the 5C state?

Response: Thanks for the reviewer's comment. Approximately 3% of E4.5 cells (1 in 29) and 25% of E5.5 cells (69 in 267) expressed both naive and primed markers above average (**Figure 7F**). In addition, several representative cells with high expression of naive and primed markers were summarized in **Figure 7G**.

minor points

1) The authors show that the 5C medium promotes HIF1a expression and suggest that HIF1a in turn regulates expression of epigenetic modifiers such as Wdr5 directly. In the absence of HIF1a ChIP data, indirect effects e.g. through HIF1a-induced TGFb expression (Fig. EV3A and Mingyuan et al. 2018) cannot be excluded and should be mentioned.

Response: Thanks for the reviewer's suggestion. In the current studies, we concluded that the five epigenetic factors can be upregulated by overexpression of *Hif1a*, oligomycin treatment, and overexpression of several mesenchymal transcriptional factors both in MEFs and during reprogramming. We can not exclude the contribution from other factors, especially when pathways related to EMT and OGS are highly connected. Additional discussion was provided in **page 29 paragraph 2**.

2) The authors have generated a remarkable amount of data. However, the description of the experimental design and obtained results is often insufficient. Describing the conducted experiments in the figure legends in greater detail and introducing key technical features of the work e.g. the GOF18ΔPE transgene and AP-staining as means to discriminate between pre-iPSCs and iPSC would be beneficial for the reader.

Response: Thanks for the reviewer's suggestion. We have provided the abovementioned information in the revised manuscript (Page 5 Paragraph 1, Page 8 Paragraph 1), and also re-written the manuscript to improve clarity.

3) Fig. 1H/1L: The measured value (likely ECAR [mpH/min]) should appear on the Y-axis

Response: Thanks for the reviewer's suggestion. We have provided these information in the figures in the revised manuscript.

4) Fig. EV1B: In addition to the merge, it would be good to show the Oct4/Nanog/Rex1 staining/channel separately. This could help to assess potential heterogeneity within the 5C colonies. Scale bars are missing in this panel.

Response: Thanks for the reviewer's suggestion. We provided the immunofluorescence images in separated channels in the revised manuscript in Appendix Figure S1B. The scale bars are also provided.

5) Fig.2H upper part: As GFP expression is the readout in this experiments the figure should state "iPSC colonies" instead of "pre-iPSC colonies"

Response: Thanks for the reviewer's suggestion. We separated the upper and lower parts into two panels in the revised manuscript. In the revised Figure 3D, we modified the schematic illustration. The pre-iPSCs isolated during reprogramming with three different media were used to be treated with different medium or factors for an additional seven days. Therefore, "pre-iPSC colonies" was used here. We have edited the legend and provided an additional description in page 11 paragraph 3.

6) Fig.3F: It should be stated in the figure legend where MEFs were transfected with OSKM simultaneously or sequentially.

Response: Thanks for the reviewer's suggestion. We have added this information in revised manuscript (page 32 paragraph 2, page 14 paragraph 1). When MEFs were transfected with OSKM sequentially, we used OK+M+S in figure and described it in legend.

7) Fig.4H: It is unclear which significance level belongs to which comparison

Response: Thanks for the reviewer's suggestion. We have made it clear in revised manuscript. The unnecessary comparisons were removed to make it clearer.

8) Fig.6F. It is unclear what is plotted here on the X- and Y-axis. A more details description in the figure legend would be beneficial

Response: Thanks for the reviewer's suggestion. We have made it clear in revised manuscript. The original Figure 6F is **Figure 7F** in revised manuscript. Briefly, markers of 5C, naïve and primed states (**Dataset EV4**) were used to analyze expression profiles of epiblast during the early development of mouse embryos (Data ref: Gene Expression Omnibus GSE100597, 2017). Single cell was plotted based on the average expression of each group of markers.

9) page 18: When the authors describe the higher expression of Hif1a/2a in 5C colonies it should be referred to Fig.EV6E instead of EV5E

Response: Thanks for the reviewer's suggestion. We have made the correction in revised manuscript (**page 25 paragraph 2**).

10) page 22: ... Gastrulation is the segregation of the "post-implantation epiblast" into three germ layers ...

Response: Thanks for the reviewer's suggestion. We have made the correction in revised manuscript (**page 30 paragraph 3**).

Referee #3:

Sun et al. investigate mechanisms by which a chemically defined medium (5C) enhances the efficiency of reprogramming of MEFs to iPSCs. In particular, they explore roles of a metabolic shift from oxidative phosphorylation to glycolysis (OGS) and epithelial-mesenchymal transition (EMT), both of which they find to be increased in 5C as compared to traditional reprogramming culture media. While OGS and EMT were previously reported to promote reprogramming, the authors now explore their interconnection and provide evidence that OGS and EMT reinforce each other and act together in inducing several epigenetic factors that accelerate reprogramming. The authors further

follow up on their previous findings that EMT inhibits late stages of reprogramming and demonstrate that OGS contributes to this phenomenon. As a consequence, high EMT and OGS trap cells in an intermediate stage that appears to have similarity to embryonic epiblast cells.

The interplay between metabolism, epigenetics and transcriptional control in shaping reprogramming efficiency and stem cell identity is a timely and interesting subject. The authors present a large amount of data that carefully document how modulating OGS and EMT alone and together affects reprogramming efficiency. The results give novel and interesting insights that help connect previously isolated findings concerning these processes. The correlation between the 5C state and epiblast cells seems rather speculative, but the progress in iPSC generation reported in the present work should be of great interest to the stem cell community.

Major points:

1. The relevance of OGS as compared to other transcriptional activities of HIF1 α is not clear. To address this, the authors should increase / inhibit glycolysis in the context of HIF1 α over-expression or knockdown and determine the effects on reprogramming in this context. Along these lines, to strengthen the experiments concerning the postulated feedback loop between HIF1 α /OGS and EMT, HIF1 α and EMT transcription factors should be manipulated together and in the context of increased/inhibited glycolysis, and reprogramming efficiency tested. In particular, is there any input from metabolism beyond the transcriptional activities of HIF1 α and the EMT transcription factors into expression of the identified epigenetic factors (5F)?

Response: Thanks for the reviewer's suggestion. In the revised manuscript, the suggested experiments were performed in (Appendix Figure S5C). *Hif1 α* expression positively correlated with reprogramming efficiency in the presence of 2-DG or oligomycin. OGS or metabolism state also positively correlated with reprogramming efficiency in the context of *Hif1 α* overexpression or knockdown (Appendix Figure S5C). Thus, both *Hif1 α* -independent OGS and the OGS-unrelated transcriptional activity of HIF1 α may contribute to reprogramming. Actually, the pathway from 5C medium to early EMT/facilitated OGS, then to up-regulated epigenetic factors, and finally to the promoted reprogramming was suggested and confirmed in the current manuscript. However, we can not or nearly impossible to exclude other potential mechanisms underlying the beneficial roles of 5C

medium. We have discussed this in the Discussion section in revised manuscript page 29 paragraph 2.

Furthermore, the expression of key mesenchymal transcriptional factors was manipulated with TGF β and Repsox (day 2-7 treatment) when *Hif1 α* expression and/or OGS were manipulated (Appendix Figure S5D). A short treatment of TGF β consistently promoted reprogramming. Repsox treatment promoted reprogramming in the presence of *sh-Hif1 α* and/or 2-DG, while impaired reprogramming in the presence of *Hif1 α* and/or oligomycin, which further explored the connection among *Hif1 α* expression, EMT, and OGS. However, because modulating one of the three aspects (*Hif1 α* expression, EMT, and OGS) may affect the other two, it is nearly impossible to identify a clean connection between only two of them. Additional discussion was provided in page 15 paragraph 3.

In addition, during reprogramming with mES medium, *Hif1 α* induced larger upregulation of epigenetic factors than oligomycin on day 6 (Appendix Figure S5A). Since oligomycin did not affect the expression of *Hif1 α* or *Pdk1/2* (Appendix Figure S5B), and has lower ability than *Hif1 α* to induce EMT transcriptional factor (Figure 4A-C & EV2B), metabolic state contributes to the regulation of epigenetic factors independently of *Hif1 α* and EMT transcriptional factors (page 15 paragraph 2).

2. In Figure 2, the authors identify 5 epigenetic factors with diverse activities / functions that are upregulated in 5C medium and contribute to reprogramming efficiency. I understand that the present work focuses on improving reprogramming protocols, but it would be helpful to have at least some indication that the observed changes in 5F levels result in epigenetic changes and altered expression of pluripotency factors.

Response: Thanks for the reviewer's suggestion. When these five factors were overexpressed simultaneously during reprogramming with mES medium, an increase of Oct4-GFP⁺ colonies on day 15 and upregulation of pluripotency markers on day 6 was observed (Figure 2E-F). In addition, consistent epigenetic changes at several core pluripotency loci were also observed on day 6 with ChIP-qPCR (Figure 2G). Moreover, Oct4 is sufficient to help these five factors induce similar epigenetic and expression changes (Figure 2E-G). Therefore, 5C medium promotes reprogramming by upregulating these five epigenetic factors at least partially (page 9 paragraph 3).

3. The authors conclude that 5F function downstream of EMT and OGS - does this mean that they function independently of metabolic state, if they are expressed? This is of particular interest in the context of the present study, because metabolite levels can influence the abundance of epigenetic marks directly, independently of altered transcription of the enzymes that add / erase these marks.

Response: Thanks for the reviewer's suggestion. During reprogramming with mES medium, Hif1 α induced larger upregulation of epigenetic factors than oligomycin on day 6 (Appendix Figure S5A). Since oligomycin did not affect the expression of Hif1 α or Pdk1/2 (Appendix Figure S5B), and had lower ability than Hif1 α to induce EMT transcriptional factors (Figure 4A-C & EV2B), metabolic state contributes to the regulation of epigenetic factors independently of Hif1 α and EMT transcriptional factors.

Minor points:

1. p. 12 Figure S4F should be S3F

Response: We are sorry to make the mistake. We have made corrections in the revised manuscript.

Additional suggestions:

The readability of the manuscript could be improved by discussing the data in more detail and separating data presentation from interpretation more clearly. Two examples to make this point:

- p. 6, Figure 1C-D: '5C medium induced a larger metabolic switch from oxidative phosphorylation to glycolysis than mES medium'. Figure 1C shows a clear increase in ECAR, whereas Figure 1D shows but a minor decrease in OCR, suggesting that glycolysis is increased whereas oxidative phosphorylation is largely unperturbed. The measured parameters should be explicitly referred to and the obtained results discussed separately.

- p. 12, Figure 3D: 'These methods modulated OGS similarly to HIF1 α or sh-HIF1 α but induced EMT to a lesser extent.' The figure shows a quantification of cell migration as a proxy for EMT, which should be made clear in the text.

Response: We are sorry to make the mistake. We have re-written the manuscript to improve clarity.

We have discussed the ECAR and OCR results separately in Figure 1C-D (revised Figure 1D-E) (page 6 paragraph 4) as suggested by the reviewer. In revised Figure 1E, the difference in

OXPPOS is difficult to be labeled. However, in Appendix Figure S2C-F, the results of statistical analysis of ECAR and OCR studies in cells on day 6 during reprogramming with mES and 5C medium were provided. 5C medium significantly increased glycolysis and inhibited OXPPOS.

For using cell migration to quantify EMT or MET, we have discussed this at the first time when we mentioned this studies (page 5 paragraph 3), thus we did not mention it again in original Figure 3D (revised Figure 4B).

Reference

Chen J, Liu H, Liu J, Qi J, Wei B, Yang J, Liang H, Chen Y, Chen J, Wu Y, Guo L, Zhu J, Zhao X, Peng T, Zhang Y, Chen S, Li X, Li D, Wang T, Pei D (2012) H3K9 methylation is a barrier during somatic cell reprogramming into iPSCs. *Nature genetics* 45: 34-42

Kalkan T, Olova N, Roode M, Mulas C, Lee HJ, Nett I, Marks H, Walker R, Stunnenberg HG, Lilley KS, Nichols J, Reik W, Bertone P, Smith A (2017) Tracking the embryonic stem cell transition from ground state pluripotency. *Development* 144: 1221-1234

Lin L, Liang L, Yang X, Sun H, Li Y, Pei D, Zheng H (2018) The homeobox transcription factor MSX2 partially mediates the effects of bone morphogenetic protein 4 (BMP4) on somatic cell reprogramming. *The Journal of biological chemistry* 293: 14905-14915

Liu X, Sun H, Qi J, Wang L, He S, Liu J, Feng C, Chen C, Li W, Guo Y, Qin D, Pan G, Chen J, Pei D, Zheng H (2013) Sequential introduction of reprogramming factors reveals a time-sensitive requirement for individual factors and a sequential EMT-MET mechanism for optimal reprogramming. *Nat Cell Biol* 15: 829-838

Mikkelsen TS, Hanna J, Zhang X, Ku M, Wernig M, Schorderet P, Bernstein BE, Jaenisch R, Lander ES, Meissner A (2008) Dissecting direct reprogramming through integrative genomic analysis. *Nature* 454: 49-55

Rostovskaya M, Stirparo GG, Smith A (2019) Capacitation of human naive pluripotent stem cells for multi-lineage differentiation. *Development* 146

Samavarchi-Tehrani P, Golipour A, David L, Sung HK, Beyer TA, Datti A, Woltjen K, Nagy A, Wrana JL (2010) Functional genomics reveals a BMP-driven mesenchymal-to-epithelial transition in the initiation of somatic cell reprogramming. *Cell Stem Cell* 7: 64-77

2nd Editorial Decision

13th Jan 2020

Thank you for submitting your revised manuscript for consideration by The EMBO Journal. Please accept my apologies for the unusual delay in processing your revised manuscript due to protracted referee input. Your amended study was sent back to three of the referees for re-evaluation, and we have received comments from two of them, which I enclose below. Please note that while referee #1 was not able to look back into your complemented work at this time, we have editorially assessed your rebuttal comments and found his/her concerns to be satisfactorily addressed.

As you will see the other referees find that their concerns have been sufficiently addressed and they are now in favour of publication.

Thus, we are pleased to inform you that your manuscript has been accepted in principle for publication in The EMBO Journal, pending the remaining minor issues are addressed by additional experiments or introducing caveats where appropriate.

Also, we need you to consider some issues related to formatting and data representation as listed below, which need to be adjusted at re-submission.

REFeree REPORTS:

Referee #2:

The revised manuscript is greatly improved in both clarity of writing and experimental evidence supporting the author's model. The single cell qPCR analysis of the 5C cells and the identification of cells in the peri-implantation embryo that share characteristics with 5C cells convincingly support the proposed 5C pluripotent state, which is distinct from naive and prime pluripotency. In general, the additional experimental evidence and clarifications provided by the authors addressed the major points I have raised convincingly.

However, prior to publication, improvement is required in a few parts of the manuscript describing data processing and experimental methods.

Minor points:

1. When the authors investigated iPSC colonies generated using the "5C-Primed" approach (EV5A), how were primed iPSC colonies distinguished and from the remaining pre-iPSC colonies e.g. in the case of the E-Oct4GFP- colonies?
2. Fig1F/H and Fig2D: The authors used shRNAs to knock down Pdk1/2, Hif1a, and Bmi1 ...etc. The knock-down efficiencies and the shRNA sequences should be provided.
3. EV1D: Which thresholds were used to group the cells in the different migration abilities? What is the 'control' shown in this panel?
4. Fig7C, D,E: It is not clear how values that are depicted here were generated. Are these Z scores of the expression in Log2 FPKM?
5. Fig7F, I and EV3G: Here, the average expression of different marker gene groups in single cells are plotted. It would be beneficial to indicate high and low average expression on both axes.
6. In some figures panels appear to be incompletely labelled or swapped.
 - Fig 2D: Likely the "5C " and " mES" column labels are missing
 - Fig 5I: Labels on the X-axis seem to be swapped.
 - Fig 7A and B: Panels seem to be swapped compared to the description on page 26.
 - Fig 7D: Labels of the columns in the right panel are missing
 - Appendix Fig S7 D and E: It is possible that the Y-axis labels are incomplete. While the left diagram likely shows "Reprogramming with mES medium", the right diagram depicts "Reprogramming with 5C medium".

Referee #3:

In the revised manuscript, Sun et al. perform experiments in which HIF1a, and OGS are activated or inhibited alone or in combination. The results clearly show these processes affect reprogramming efficiency by themselves and also have additive effects. The authors further demonstrate epigenetic changes induced by F5, which correlate with the observed transcriptional changes in pluripotency factors.

As I pointed out in my previous review, to support the authors model that 5F functions downstream of OGS it would be good to conduct similar experiments to establish whether 5F cooperate with metabolic state to increase reprogramming efficiency, or whether they function independently, once expressed (e.g. overexpression of 5F together with oligomycin or 2-DG treatment).

In general, the new data support the author's conclusions and with the additional experiment suggested above, I would recommend publication in the EMBO Journal.

2nd Revision - authors' response

21st Jan 2020

Referee #2:

The revised manuscript is greatly improved in both clarity of writing and experimental evidence supporting the author's model. The single cell qPCR analysis of the 5C cells and the identification of cells in the peri-implantation embryo that share characteristics with 5C cells convincingly support the proposed 5C pluripotent state, which is distinct from naive and prime pluripotency. In general, the additional experimental evidence and clarifications provided by the authors addressed the major points I have raised convincingly.

However, prior to publication, improvement is required in a few parts of the manuscript describing data processing and experimental methods.

Minor points:

1. When the authors investigated iPSC colonies generated using the "5C-Primed" approach (EV5A), how were primed iPSC colonies distinguished and from the remaining pre-iPSC colonies e.g. in the case of the E-Oct4GFP- colonies?

Response: Thanks for the review's comment.

We used GFP fluorescence and cell morphology to distinguish different types of colonies. Type A-Oct4GFP+ and -Oct4GFP- colonies had relatively compact morphology and were conventional colonies induced with mES medium, while type B-Oct4GFP+ and -Oct4GFP- colonies had loose morphology and were typical colonies induced with 5C medium. Type C-Oct4GFP+ and type E-Oct4GFP- colonies had morphologies typical to those of naive (compact, domed and Oct4GFP+) and primed PSCs (monolayer without Oct4GFP), respectively. Type D-Oct4GFP- colonies were even looser than type B and negative for Oct4GFP fluorescence (Page 19 Paragraph 4).

We considered primed colonies as type E-Oct4GFP- colonies. These colonies (monolayer without Oct4GFP) were distinguished from the other pre-iPSCs colonies by morphology. The other pre-iPSCs colonies (type A-, B-, and D-Oct4GFP- colonies) are domed.

2. Fig1F/H and Fig2D: The authors used shRNAs to knock down Pdk1/2, Hif1a, and Bmi1 ...etc. The knock-down efficiencies and the shRNA sequences should be provided.

Response: Thanks for the review's comment.

The abilities of sh-RNAs (sh-Hif1 α , sh-Pdk1, and sh-Pdk2) to suppress the expression of their target genes were determined three days after being delivered into MEFs (Appendix Figure S2C).

The abilities of sh-RNAs (sh-Bmi1, sh-Ctcf, sh-Ezh2, sh-Kdm2b and sh-Wdr5) to suppress the expression of their target genes were determined on day 6 during reprogramming with 5C medium (Appendix Figure S3D).

The abilities of sh-RNAs (sh-Snai2, sh-Twist1, sh-Twist2, and sh-Zeb1) to suppress the expression of their target genes were determined three days after being delivered into MEFs (Appendix Figure S3E).

The shRNA sequences were provided in Dataset EV6.

3. EV1D: Which thresholds were used to group the cells in the different migration abilities? What is the 'control' shown in this panel?

Response: Thanks for the review's comment.

Cell migration was determined on day 3 during reprogramming with live-cell imaging. One third of cells with the highest/lowest migration abilities were separated into high/low group. The other one third cells were separated into medium group. Reprogramming of these cells were traced via live-cell imaging. The number of Oct4GFP⁺ colonies on day 15 converted from different groups of cells was summarized. "Control" was the average of Oct4GFP⁺ colonies generated from these three groups of cells. We have added more discussion in legend of Figure EV1D.

4. Fig7C, D,E: It is not clear how values that are depicted here were generated. Are these Z scores of the expression in Log₂ FPKM?

Response: Thanks for the review's comment.

> Figure 7C. Epithelial, mesenchymal, glycolysis and oxidative phosphorylation (OXPHOS) markers were listed in Dataset EV4.

> Figure 7D. Genes with higher expression at different developmental stages were selected out from previously reported RNA-seq (Data ref: Bertone, 2015).

> Figure 7E. Markers of 5C, naïve and primed states were selected based on the current RNA-Seq (Dataset EV4).

For each marker, Log₂ values of FPKM were normalized by subtracting the average of four (left panel) or five groups (right panel). Log₂ values of FPKM of each group of markers were then averaged and plotted.

Additional discussion as provided in the legend of Figure 7C-E.

5. Fig7F, I and EV3G: Here, the average expression of different marker gene groups in single cells are plotted. It would be beneficial to indicate high and low average expression on both axes.

Response: Thanks for the review's comment. We have added "high" and "low" on both axes.

6. In some figures panels appear to be incompletely labelled or swapped.

• Fig 2D: Likely the "5C " and " mES" column labels are missing

Response: Thanks for the review's suggestion. We have added "5C medium" and "mES medium" to Figure 2D.

- Fig 5I: Labels on the X-axis seem to be swapped.

Response: Thanks for the review's suggestion. We have correct the labels of the first and the third bars in Figure 5I.

- Fig 7A and B: Panels seem to be swapped compared to the description on page 26.

Response: Thanks for the review's suggestion. We have switched Figure 7A with 7B.

- Fig 7D: Labels of the columns in the right panel are missing

Response: Thanks for the review's suggestion. We have added the labels of the columns in the right panel in Figure 7D.

- Appendix Fig S7 D and E: It is possible that the Y-axis labels are incomplete. While the left diagram likely shows "Reprogramming with mES medium", the right diagram depicts "Reprogramming with mES medium"

Response: Thanks for the review's suggestion. We have corrected the labels "Reprogramming with mES medium" and "Reprogramming with mES medium" in Appendix Figure S7D and E. In both Figure S7D and S7E, "Reprogramming with mES medium" was in the left panel and "Reprogramming with mES medium" was in the right panel.

Referee #3:

In the revised manuscript, Sun et al. perform experiments in which HIF1a, and OGS are activated or inhibited alone or in combination. The results clearly show these processes affect reprogramming efficiency by themselves and also have additive effects. The authors further demonstrate epigenetic changes induced by F5, which correlate with the observed transcriptional changes in pluripotency factors.

As I pointed out in my previous review, to support the authors model that 5F functions downstream of OGS it would be good to conduct similar experiments to establish whether 5F cooperate with metabolic state to increase reprogramming efficiency, or whether they function independently, once expressed (e.g. overexpression of 5F together with oligomycin or 2-DG treatment).

In general, the new data support the author's conclusions and with the additional experiment suggested above, I would recommend publication in the EMBO Journal.

Response: Thanks for the review's suggestion.

The expression of *Bmi1*, *Ctcf*, *Ezh2*, *Kdm2b*, and *Wdr5* was modulated with overexpression or sh-RNA-mediated knockdown via a retrovirus system. Energy metabolism was directly controlled by using small-molecule compounds, oligomycin (1 μ M) and 2-DG (5 mM). The expression of *Bmi1*, *Ctcf*, *Ezh2*, *Kdm2b*, and *Wdr5* positively correlated with reprogramming efficiency in the presence of 2-DG or oligomycin (Appendix Figure S5F). Thus, these five factors affect reprogramming efficiency by themselves and also have additive effects. (Page 16 Paragraph 2).

Overexpression of *Ctcf*, *Kdm2b*, and *Wdr5* (3F) did not affect energy metabolism and cell migration (Figure 3H-K), which confirmed that the upregulation of these epigenetic factors was downstream of OGS and EMT.

Major Changes

- 1) Add new data as Appendix Figure S2C, S3D-E, and S5F;
- 2) Provide new discussion to address the reviews' comments;
- 3) Correct the mistakes which were pointed out by the reviewer.
- 4) Format the manuscript to meet the requirements of EMBO J.
- 5) Proofread the manuscript again.

The revised manuscript were provided in two versions, clear version (all changed accepted) and edited version (change-tracked mode).

3rd Editorial Decision

23rd Jan 2020

Thank you for submitting the revised version of your manuscript. I have now evaluated your amended manuscript and concluded that the remaining minor concerns have been sufficiently addressed.

Thus, I am pleased to inform you that your manuscript has been accepted for publication in the EMBO Journal.

Corresponding Author Name: Hui Zheng & Duanqing Pei

Journal Submitted to: EMBO J

Manuscript Number: EMBOJ-2019-102961R